## METHOD

# siVAE: interpretable deep generative models for single-cell transcriptomes

Yongin Choi[1,2], Ruoxin Li[2,3] and Gerald Quon[1,2,4*]

*Correspondence:
gquon@ucdavis.edu

[1] Graduate Group in Biomedical Engineering, University of California, Davis, Davis, CA, USA
[2] Genome Center, University of California, Davis, Davis, CA, USA
[3] Graduate Group in Biostatistics, University of California, Davis, CA, USA
[4] Department of Molecular and Cellular Biology, University of California, Davis, Davis, CA, USA

**Abstract**

Neural networks such as variational autoencoders (VAE) perform dimensionality reduction for the visualization and analysis of genomic data, but are limited in their interpretability: it is unknown which data features are represented by each embedding dimension. We present siVAE, a VAE that is interpretable by design, thereby enhancing downstream analysis tasks. Through interpretation, siVAE also identifies gene modules and hubs without explicit gene network inference. We use siVAE to identify gene modules whose connectivity is associated with diverse phenotypes such as iPSC neuronal differentiation efficiency and dementia, showcasing the wide applicability of interpretable generative models for genomic data analysis.

## Introduction

Single-cell genomic assays such as scRNA-seq and scATAC-seq measure the activity level of tens to hundreds of thousands of genomic features (genes or genomic regions), yielding high dimensional measurements of individual cells. Features tend to be inter-correlated: gene members of the same pathway, complex, or module exhibit correlated expression patterns across cells [1], and proximal genomic regions covering the same regulatory elements or expressed genes are correlated in their accessibility patterns [2]. Common downstream analysis tasks such as visualization [3], clustering [4], trajectory inference [5, 6], and rare cell type identification [7, 8] typically do not directly compute on the original features. Instead, they first perform dimensionality reduction (DR) to project cells from their high-dimensional feature space to a lower-dimensional cell-embedding space consisting of a smaller set of embedding dimensions. Individual embedding dimensions capture distinct groups of correlated input features and are often also correlated with biological factors such as case-control status [9], gender [10], and others [11]. Downstream tasks are then carried out on these embedding dimensions.

Given the central role of embedding dimensions in analysis, it is useful to be able to characterize and interpret which of the original input features contributed to the construction of each embedding dimension. For example, in a visualization of a 2D

cell-embedding space, interpretation of the embedding dimensions would identify genes that explain variation in the transcriptome along different axes (Fig. 1). Linear DR frameworks such as PCA achieve interpretation through estimation of the contribution of individual features towards each embedding dimension. However, linear DR frameworks are less powerful because they can often be viewed as restricted implementations of non-linear frameworks [12]. In contrast, non-linear methods such as UMAP, t-SNE, and variational autoencoders (VAEs) produce better visualizations in which cells of the same cell type cluster together more closely (Additional file 1: Fig. S1), but they are not interpretable and require more ad hoc, downstream analysis to gain intuition about the factors driving the arrangement of cells in the visualization. Beyond visualization, interpretability is an important property for other tasks, such as the detection of genes and pathways driving variation in expression within or across cell types [13] and the identification of genes associated with cellular trajectories directly from visualization [14].

Previous works have explored extensions of the non-linear VAE framework to achieve interpretability. Methods such as LDVAE [15], scETM [16], and VEGA [17] achieve interpretability by imposing a linear relationship between the latent embedding layer and the output layer in the decoder, which in turn makes these approaches effectively linear dimensionality reduction methods. Other ad-hoc approaches such as DeepT2Vec

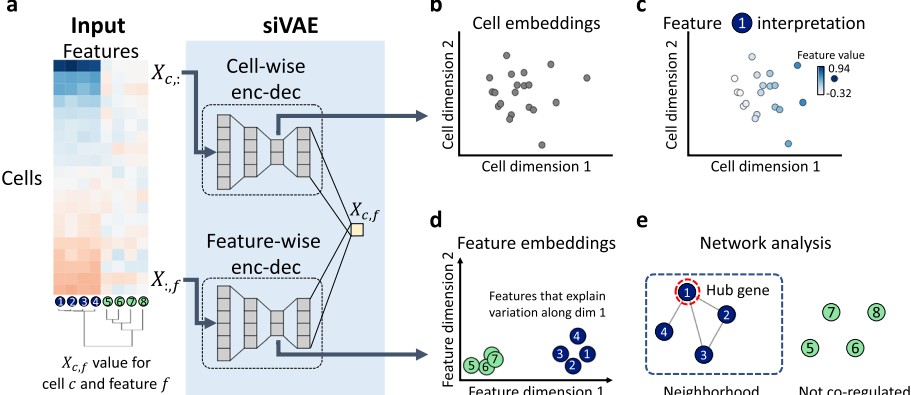

**Fig. 1** siVAE infers interpretable representations of single-cell genomic data. **a** The input to siVAE is a cell-by-feature matrix; shown here is a synthetic gene expression matrix of eight genes, four of which are tightly regulated (genes 1–4), and the other four of which vary independently (genes 5–8). siVAE is a neural network consisting of a pair of encoder-decoders, that jointly learn a cell-embedding space and feature embedding space. The cell-wise encoder-decoder acts similarly to a canonical VAE, where the input to the encoder is a single cell $c$'s measurement across all input features ($X_{c,:}$). The cell-wise encoder uses the input cell measurements to compute an approximate posterior distribution over the location of the cell in the cell-embedding space. The feature-wise encoder-decoder takes as input measurements for a single feature $f$ across all input training cells ($X_{:,f}$), and computes an approximate posterior distribution over the location of the feature in the feature embedding space. The decoders of the cell-wise and feature-wise encoder-decoders combine to output the expression level of feature $f$ in cell $c$ ($X_{c,f}$). **b,d** Visualization of the cell and feature embedding spaces learned from the gene expression matrix in **a**. Note in **d** that the embeddings of genes 1, 2, 3, and 4 all have large magnitudes along dimension 1 but not dimension 2, suggesting genes 1, 2, 3, and 4 explain variation in the cell-embedding space along dimension 1. Genes 5, 6, 7, and 8 are located at the origin of the feature embedding space, suggesting they do not co-vary with other features. **c** The expression patterns of gene 1 are overlaid on the cells in the cell-embedding space. Gene 1 clearly increases in expression when inspecting cells from left to right, consistent with the feature embedding space that shows Gene 1 having large loadings on dimension 1. **e** A trained siVAE model can be used to identify hubs and gene neighbors in a gene co-expression network, without the need to explicitly infer a co-expression network

[18] and deepAE [19] combine unsupervised autoencoders with a supervised loss function that leverages prior knowledge such as cell type identity, thus only yielding insight with respect to those pre-defined features.

Here we propose a scalable, interpretable variational autoencoder (siVAE) that combines the non-linear DR framework of variational autoencoders with the interpretability of linear PCA. siVAE is a variant of VAEs that additionally infers a feature embedding space for the genomic features (genes or genomic regions) that is used to interpret the cell-embedding space. Importantly, by using a non-linear network to combine the cell and feature embedding space, siVAE achieves interpretability without introducing linear constraints, making it strictly more expressive than LDVAE, scETM, and VEGA. Compared to other approaches for achieving interpretable, non-linear DR, siVAE is either faster, generates more accurate low-dimensional representations of cells, or more accurately interprets the non-linear DR without introducing linear restrictions or dependence on prior knowledge.

## Results

siVAE is a deep neural network consisting of two pairs of encoder-decoder structures, one for cells and the other for features (Fig. 1a). The cell-wise encoder-decoder learns to compress per-cell measurements $X_{c,:}$ (where $X$ is a matrix of dimension $C \times F$, $C$ indexes cells, and $F$ indexes features) into a low-dimensional embedding ($z_c$) of length $K$ for visualization and analysis, similar to traditional VAEs implemented in single-cell genomic applications and others [20–22]. We call the $C \times K$ matrix of embeddings $Z$ the siVAE score matrix, where the scores of cell $c$ ($Z_{c,:}$) represent its position in the cell-embedding space.

To facilitate interpretation of the cell-embedding space, siVAE additionally implements a separate feature-wise encoder-decoder network (Fig. 1a) that learns to compress per-genomic features across the training cells ($X_{:,f}$) into a low-dimensional embedding ($v_f$) of length $K$, analogous to the cell-wise encoder-decoder. We call the $F \times K$ matrix of feature embeddings $V$ the siVAE loading matrix, where the loadings of feature $f$ ($V_{f,:}$) represent its position in the feature embedding space. The cell- and feature-wise decoders together are used to generate the observed measurement $X_{c,f}$. The strategy siVAE uses to achieve interpretation is best understood by briefly reviewing why probabilistic PCA (PPCA) and factor analysis are interpretable [15, 23]. The underlying generative model behind PPCA can be thought of as similar to a VAE with a linear decoder, and the output of PPCA includes both a factor loading matrix $V$ and score matrix $Z$. In probabilistic PCA, the mean predicted expression of feature $f$ in cell $c$ ($X_{c,f}$) is assumed to be $V_{f,:}^T Z_{c,:}$, the dot product of the loadings for feature $f$ and the scores of cell $c$. PPCA is therefore interpretable, because the larger the contribution of a feature $f$ to a particular dimension $k$ (indicated by the magnitude of $V_{f,k}$), the more the measurement of feature $f$ ($X_{c,f}$) is influenced by a cell's corresponding score in that dimension ($Z_{c,k}$). Conversely, when the magnitude of $V_{f,k}$ is small (or even 0), then the cell's corresponding score in that dimension ($Z_{c,k}$) does not influence $X_{c,f}$, the measurement of feature $f$ in cell $c$. In this regard, we say that the PPCA model enforces correspondence between $Z_{c,k}$ and $V_{f,k}$, along the dimension $k$ of both the cell and feature embeddings.

siVAE achieves interpretability of the siVAE scores $Z_{c,k}$ by adding a small interpretability regularization term to its objective function (see Methods). More specifically, this regularization term penalizes deviation between the observed measurement $X_{c,f}$ and the dot product of the corresponding siVAE scores and loadings ($V_{f,:}^{T} Z_{c,:}$). This small regularization term helps enforce soft correspondence between dimension $k$ of the cell scores, and dimension $k$ of the feature loadings.

Our framework for making VAEs interpretable is generalizable to other VAE-based frameworks. Given that VAEs have been applied to a wide range of genomics data modalities (epigenomics [24–26] and miRNA [27]) and analysis (visualization [20, 28], trajectory inference [29], data imputation [30], and perturbation response prediction [31–33]), our work can therefore enable interpretability in a wide range of downstream applications of VAEs.

### Results – siVAE accurately generates low-dimensional embeddings of cells

We first evaluated siVAE in the context of cell-embedding space inference, where the goal is to generate low-dimensional representations of cells in which cells of the same cell type cluster together. We benchmarked siVAE against other interpretable and non-interpretable dimensionality reduction approaches using a fetal liver atlas [34] consisting of 177,376 cells covering 40 cell types. We measured the accuracy of each approach in a 5-fold stratified cross-validation framework by first using the training folds to learn a cell-embedding space, followed by training of a $k$-NN ($k=80$) classifier using the known cell type labels and cell coordinates within the embedding space. We then passed cells in the validation fold through the trained cell encoder to generate validation cell-embeddings, which were used to classify the validation fold cells. We associate higher $k$-NN accuracy with a more accurate cell-embedding space in which cells of the same type cluster together.

We compared siVAE against a (canonical) VAE as well as LDVAE [15] and scETM [16], where all four VAE frameworks used cell-wise encoder-decoders of the same size with a latent dimension of 2, and the VAE and siVAE use the same activation functions. Overall, we found siVAE's cell-embedding space to be comparable in accuracy to VAEs, suggesting that the introduction of the siVAE feature-wise encoder-decoder does not affect siVAE performance in terms of its cell-embedding space. 2D visualization of siVAE's cell-embedding space reveals striking similarity to the cell-embedding space of the VAE in that cells of the same type cluster together (Fig. 2a). Furthermore, siVAE is competitive in balanced classification accuracy with a VAE on the entire fetal liver cell atlas (Fig. 2b, Additional file 1: Fig. S2) as well as for each cell type individually (Additional file 1: Fig. S3). siVAE therefore is competitive with VAEs in terms of generating cell-embedding spaces, but has the additional benefit of interpretability, which we will explore below. In comparison, the LDVAE and scETM approach, which is interpretable like siVAE but performs linear DR, yields significantly lower classification accuracy (Fig. 2b) and generates visualizations in which different cell types mix together more prominently compared to the VAE and siVAE (Fig. 2a). LDVAE and scETM therefore gain interpretability at a significant cost to the accuracy of the cell-embedding space.

We next constructed a set of model variants of siVAE in order to identify which aspects of siVAE lead to its superior performance over LDVAE (Table 1). LDVAE is

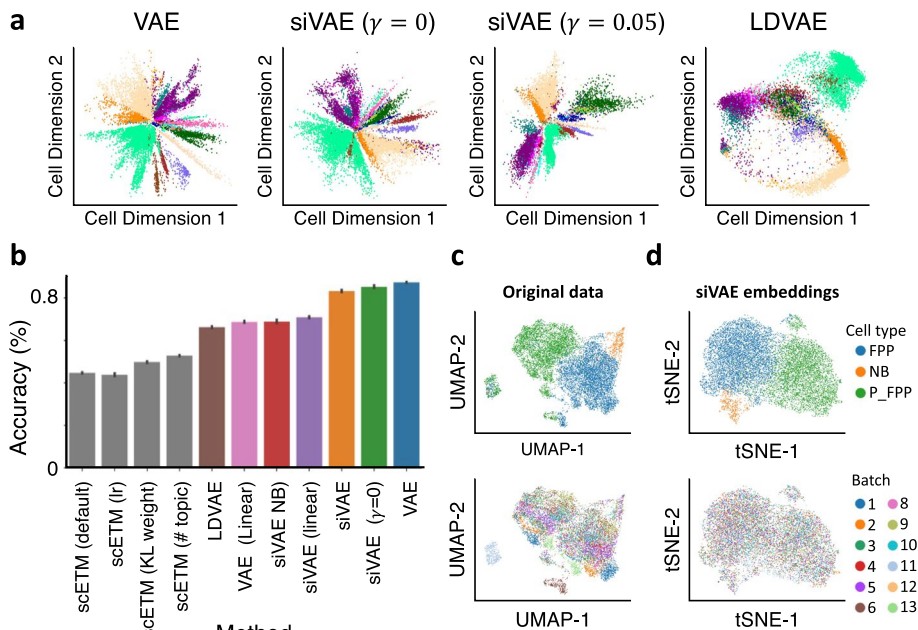

**Fig. 2** Accuracy evaluation of cell-embedding spaces. **a** 2D visualization of the inferred cell-embedding spaces of a canonical VAE, siVAE ($\gamma = 0$) (no regularization term), siVAE ($\gamma = 0.05$) (default regularization weight), and LDVAE. Each point represents a cell and is colored according to cell type. **b** Barplot indicating the balanced accuracy of a $k$-NN ($k = 80$) classifier predicting the cell type labels of single cells based on their inferred position in the cell-embedding space inferred by various methods trained on the fetal liver atlas dataset. Higher accuracies are interpreted as more accurate inferred cell-embedding spaces. **c** 2D UMAP visualization of the original NeurDiff dataset, without batch correction. Top row shows annotation based on cell type, and the bottom row shows annotation based on the batch. **d** Same as **c**, except visualization is a tSNE visualization of the siVAE-inferred cell-embedding space where siVAE corrects for batch within the model

**Table 1** List of model variants and their corresponding features. Usage of the interpretability term only applies to siVAE and its variants. A linear decoder is composed of the same number of layers as the non-linear decoder unless specified as single, in which case the latent embedding layer is directly transformed to an output layer

| Model | Interpretability Reg. | Decoder | Observation model |
|---|---|---|---|
| siVAE | Yes | Non-linear | Gaussian |
| siVAE ($\gamma = 0$) | No | Non-linear | Gaussian |
| siVAE-linear | No | Linear | Gaussian |
| siVAE-NB | Yes | Non-linear | Negative binomial |
| VAE | NA | Non-linear | Gaussian |
| scVI | NA | Non-linear | Negative binomial |
| LDVAE | NA | Linear, single | Negative binomial |
| scETM | NA | Linear, tri-factorization | Gaussian |
| VAE (linear) | NA | Linear, single | Negative binomial |

broadly similar to the classic VAE, with two key differences. First, the LDVAE decoder is restricted to use only linear activation functions in order to achieve interpretability; thus, LDVAE performs linear dimensionality reduction. Second, the LDVAE loss function uses a negative binomial or zero-inflated negative binomial distribution over the input features (genes), instead of the Gaussian distribution used in a canonical VAE. In

principle, the NB or ZINB observation model is a better fit for single-cell transcriptomic data compared to a Gaussian distribution typically used on log-transformed data [35, 36]. We therefore constructed two variants of siVAE, termed siVAE-NB and siVAE-linear. siVAE-NB is identical to siVAE, except that it uses a negative binomial distribution for the observation layer while maintaining non-linear activation functions in its decoders to achieve non-linear DR. siVAE-linear is identical to siVAE, except that it restricts both the feature-wise and cell-wise decoder to use linear activation functions like LDVAE and does not implement the interpretability term. Figure 2b and Additional file 1: Fig. S4 shows that siVAE-NB performs worse than the corresponding model with the Gaussian distribution (siVAE), suggesting that using a NB output layer does not lead to a more accurate cell-embedding space. siVAE-linear is more accurate than LDVAE (Fig. 2b), indicating that the feature-wise encoder-decoder of siVAE is overall beneficial to dimensionality reduction. However, siVAE-linear performs more poorly than siVAE, verifying the non-linear activation functions are beneficial to dimensionality reduction.

We also hypothesized that the interpretability term used in siVAE's loss function would degrade the quality of dimensionality reduction to an extent, as the interpretability term enforces correspondence between the cell and feature embeddings through a small linear dimensionality reduction penalty. We therefore constructed siVAE ($\gamma=0$), representing a siVAE model in which we turn off the regularization term by setting its weight $\gamma$ to 0, but maintain the feature embedding space. From Fig. 2b, we can see a small difference in classification performance between siVAE and siVAE ($\gamma=0$), indicating that the interpretability of siVAE comes at a small cost in classification performance. We performed additional experiments to show that when varying $\gamma$ from 0 to 100, where siVAE ($\gamma=100$) is conceptually similar to siVAE-linear. The performance of siVAE smoothly interpolates between siVAE ($\gamma=0$) to siVAE-linear's performance (Additional file 1 Fig. S5), suggesting that siVAE can smoothly balance interpretability with non-linear dimensionality reduction. In addition, we tested our model performance on imaging datasets and confirmed that both siVAE and siVAE ($\gamma=0$) showed comparable classification performance with the VAE (Additional file 1: Fig. S6,S7).

Finally, siVAE natively allows batch correction within the model similar to other approaches [16, 20]. We tested our model on an iPSC neuronal differentiation (NeurDiff) dataset [37] in which 253,381 iPSC-derived cells were sequenced using 10x Chromium before and after initiation of differentiation into neurons. We specifically focused on the samples from day 11 before differentiation, as we observed a strong batch effect with respect to pool_id (Fig. 2c). When we trained siVAE and provided batch information during training, clustering by batch is eliminated while the clustering by cell type is still preserved (Fig. 2d). These results suggest that siVAE is a viable alternative to existing dimensionality reduction approaches that can be used to perform common tasks such as dimensionality reduction, visualization and batch correction.

### Results – siVAE interprets cell-embedding spaces more accurately and faster than existing feature attribution approaches

Having shown siVAE generates cell-embedding spaces competitive with canonical VAEs, we next verified that siVAE interpretations of the embedding dimensions are accurate. Again, we define an interpretation of the cell-embedding space as a matrix of feature

loadings (or more generally, attributions) $V$ of size $F \times K$, where $F$ is the number of features (e.g., genes), $K$ is the number of cell-embedding dimensions, and the magnitude of $V_{f,k}$ indicates the strength of association between cell-embedding dimension $k$ and feature $f$ in the original data space.

In addition to methods such as siVAE and LDVAE that construct interpretable cell-embedding spaces by design, there are two types of approaches to feature attribution in the literature that can help interpret cell-embedding spaces post-inference. First, general neural network feature attribution methods can quantify the dependence of each output node of a neural network on each input node (feature) of the network [38] and includes methods such as DeepLIFT [39], saliency maps [40], grad × input [39], integrated gradients [41], Shapley value [42] and others [38, 43–47]. One of the strengths of these approaches is they can be applied to any trained neural network in principle, making them highly generalizable. Second, methods such as Gene Relevance [48] have been developed specifically to interpret cell latent spaces for any DR method, including those not based on neural networks, and can be applied after cell-embedding spaces are learned.

We first compared siVAE against Gene Relevance using the neural network feature attribution methods as a gold standard, as they have been extensively validated in other applications [49]. Figure 3a shows the mean pairwise correlation between the attributions of siVAE, Gene Relevance, and three neural net feature attribution methods (saliency maps, grad × input, and DeepLIFT). siVAE loadings are highly correlated with the neural net feature attribution methods (median Spearman $\rho = 0.73$, $P = 1.1 \times 10^{-15}$) with siVAE in striking agreement with DeepLIFT in particular (median Spearman $\rho = 0.98$, $P = 2.2 \times 10^{-16}$). In contrast, while Gene Relevance produced feature attributions that were consistent across their parameter selections (median Spearman $\rho = 0.84$, $P = 3.10 \times 10^{-22}$), they were poorly correlated with both neural net feature attribution methods (median Spearman $\rho = 0.11$, $P = 2.1 \times 10^{-6}$) and siVAE (median Spearman $\rho = 0.14$, $P = 3.9 \times 10^{-6}$). These results suggest Gene Relevance is less accurate compared to siVAE at interpreting cell-embedding spaces of VAE architectures. We also found consistent results when comparing these methods on the MNIST imaging dataset (Additional file 1: Supplementary Note 1, Additional file 1: Fig. S8).

For the above results, we applied the neural net attribution methods to the decoder of siVAE to generate the ground truth feature attributions. Previous work has suggested to apply attribution methods to the encoder to improve execution speed [19, 22]. Here we found that running attribution methods on the siVAE encoder produce substantially different interpretations that are in strong disagreement with the interpretations of the decoder (Additional file 1: Fig. S9), suggesting an interpretation of the encoder is not appropriate. These results make sense considering the primary role of the encoder is to compute an approximate posterior distribution of the latent embedding of each cell, as opposed to the decoder, which is responsible for directly mapping points from the cell-embedding space to the original data feature space. Our results therefore suggest feature attributions should be applied to the decoder of VAEs instead of the encoder.

During our experiments on interpreting cell-embedding spaces, it became evident that a number of neural network feature attribution approaches were computationally expensive to execute. Because these feature attribution methods perform calculations

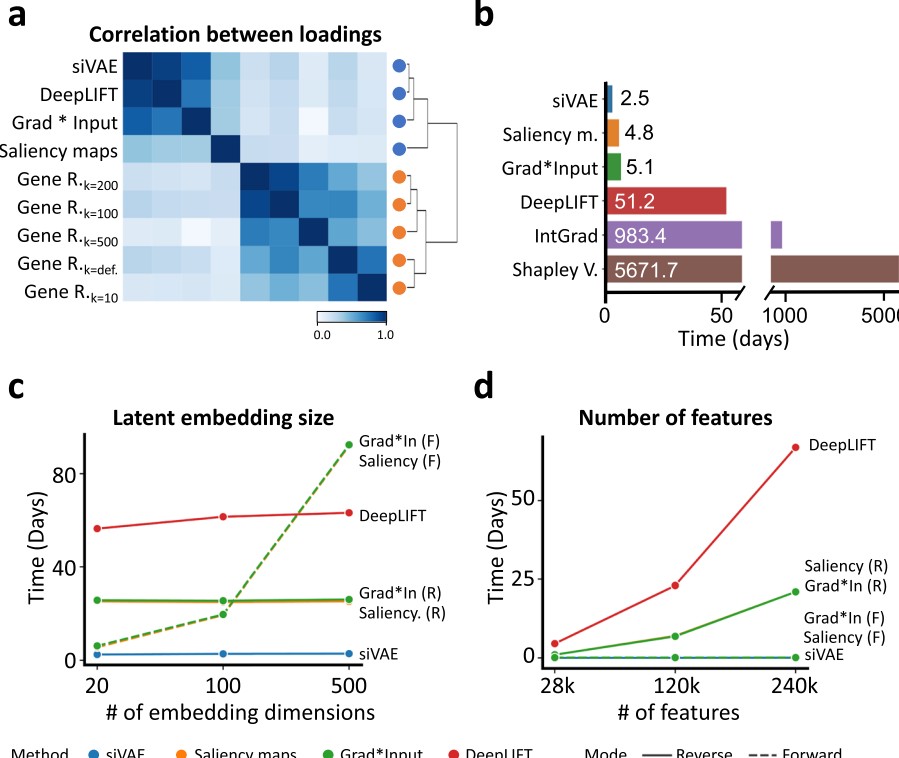

**Fig. 3** siVAE generates accurate and fast interpretations. **a** Heatmap indicating the mean pairwise correlation between the interpretations (loadings) of siVAE, Gene Relevance, and three neural net feature attribution methods (saliency maps, grad × input, and DeepLIFT), where correlations have been averaged over each of the 2 embedding dimensions for the fetal liver atlas dataset. **b** Bar plot indicating the time required to train siVAE, compared to training a canonical VAE and executing one of the feature attribution methods on the LargeBrainAtlas dataset. For feature attribution methods, the run times were extrapolated due to infeasibly long run times; additional experiments demonstrate the feasibility of extrapolation (Additional file 1: Figure S10). **c** Line plot indicating the time required to train siVAE compared to training a canonical VAE and executing a feature attribution method on the LargeBrainAtlas dataset, when the number of embedding dimensions for siVAE is varied and the number of features is fixed at 28k. **d** Line plot indicating the time required to train siVAE compared to training a canonical VAE and executing a feature attribution method on the BrainCortex dataset, when the number of features is being varied and the number of embedding dimensions is fixed at 20

separately for either each embedding dimension or each output node of the network, their run time scales linearly with the number of embedding dimensions or features when run on VAE decoders [20]. We confirmed this at a smaller scale by checking that the execution time of each batch is invariant over time (Additional file 1: Fig. S10). We reasoned that execution time will become more important in the future as the number of embedding dimensions is expected to be larger as the number of cells in the dataset grows, to accommodate more heterogeneity in the dataset. Also, the number of features would be expected to be large for assays such as scATAC-seq that profile hundreds of thousands of genomic regions or more.

We therefore hypothesized that siVAE scales faster than the neural network attribution methods on larger single-cell genomics datasets. To test this hypothesis, we assembled two datasets for execution time testing: the LargeBrainAtlas dataset published by 10x Genomics [50] consisting of 1.3 million brain cells and 27,998 genes measured with scRNA-seq, and the BrainCortex dataset [51] consisting of 8k cells and 244,544 genomic

regions measured with SNARE-seq. We first compared the execution time of training siVAE on the full LargeBrainAtlas dataset, against the run time of training a VAE and individually running one of five neural network attribution methods (saliency maps, grad × input, DeepLIFT, integrated gradients, and Shapley values) on the trained VAE. We found that siVAE achieved an execution time of 2.5 days, approximately half of the time taken by the fastest neural network attribution method (forward mode of saliency maps) (Fig. 3b).

To identify the most time-consuming step of feature attribution calculations for each method, we selected a subset of the LargeBrainAtlas dataset for varying the number of embedding dimensions from 20 to 512, and a subset of the BrainCortex dataset for varying the number of features from 28k to 240k. siVAE averaged 0.0073 days per embedding dimension (Fig. 3c) and 0.0027 days per 10k features (Fig. 3d), indicating siVAE execution time was robust to both the number of cells and features. On the other hand, we found the neural network attribution methods scale well when either the number of embedding dimensions or the number of input features is large, but not when they are both large. For example, DeepLIFT, grad ×input (reverse-mode), and saliency maps executed at 0.014, 0.0053, and 0.0012 days per embedding dimension respectively (Fig. 3c) but scaled poorly with respect to number of input features and executed at 2.9, 0.95, and 0.94 days per 10k features respectively (Fig. 3d). Switching grad × input and saliency maps to forward-mode led to fast execution times with respect to the number of input features ($5.3 \times 10^{-4}$ and $6.5 \times 10^{-4}$ days per 10k features, respectively) (Fig. 3d) but led to poor scaling with respect to the number of embedding dimensions (0.18 and 0.17 days per embedding dimension, respectively) (Fig. 3c). Slower attribution methods such as Integrated Gradients and Shapley Value were excluded due to their infeasible execution times. In summary, the neural network attribution methods scale poorly either with the number of embedding dimensions or the number of input features, depending on whether forward- or reverse-mode is used. This therefore makes their execution time slow relative to siVAE if both the number of features and embedding dimensions are large, a scenario which we expect to occur increasingly often with the ever-growing single-cell datasets.

### Results – co-expressed genes cluster in the feature embedding space

Loadings of linear DR methods such as PCA have been exploited extensively in the literature to gain insight into the structure of gene co-expression networks (GCN) [15, 23, 52]. Here, we explore the extent to which the siVAE loading matrix can be leveraged to gain insight into GCN structure. GCNs are graphs in which nodes represent genes and edges represent co-expression of a pair of genes. A GCN captures co-variation in gene expression measurements between pairs (or more) of genes across a population of cells. GCNs are of interest because they can be used to identify (1) cell population-specific gene modules, representing groups of genes that are highly co-expressed and therefore are likely to function together in a cell type-specific manner, as well as (2) gene hubs, which are genes that are connected to an unusually large number of genes (high degree centrality), and typically represent key functional genes in the cell [53, 54]. While GCN inference is valuable for interrogating gene regulatory patterns in a cell, GCN inference is a notoriously difficult and error-prone task [55–57].

Because siVAE input data is centered and scaled uniformly across all features, siVAE is forced to learn patterns of co-variation among the input features that allow accurate reconstruction of the input data from low-dimensional representations. It is therefore natural to ask whether a trained siVAE model could yield insight into the gene co-expression network structure of the training data, without the need for explicit gene network inference.

Previous work has shown that eigengenes (genes captured by PCA loadings) represent network modules in the gene co-expression network [58, 59]. We hypothesized that siVAE genes captured by feature loadings of siVAE may also represent network modules, and furthermore that co-expressed genes in the training data are proximal in the siVAE feature embedding space. To explore how groups of co-expressed genes are organized in the feature embedding space, we constructed a synthetic GCN consisting of five communities of 50 tightly correlated genes each, as well as an additional group of 50 disconnected genes (Fig. 4a). Each community follows a hub-and-spoke model in which a hub gene is connected to every other gene in the community, and each gene in the community is in turn only connected to the hub. No edges connect genes from different communities. Based on this gene network, we sampled a single-cell gene expression dataset consisting of 5000 cells and 300 genes (see Methods). The sampled expression matrix was used to train siVAE to embed genes in its feature embedding space.

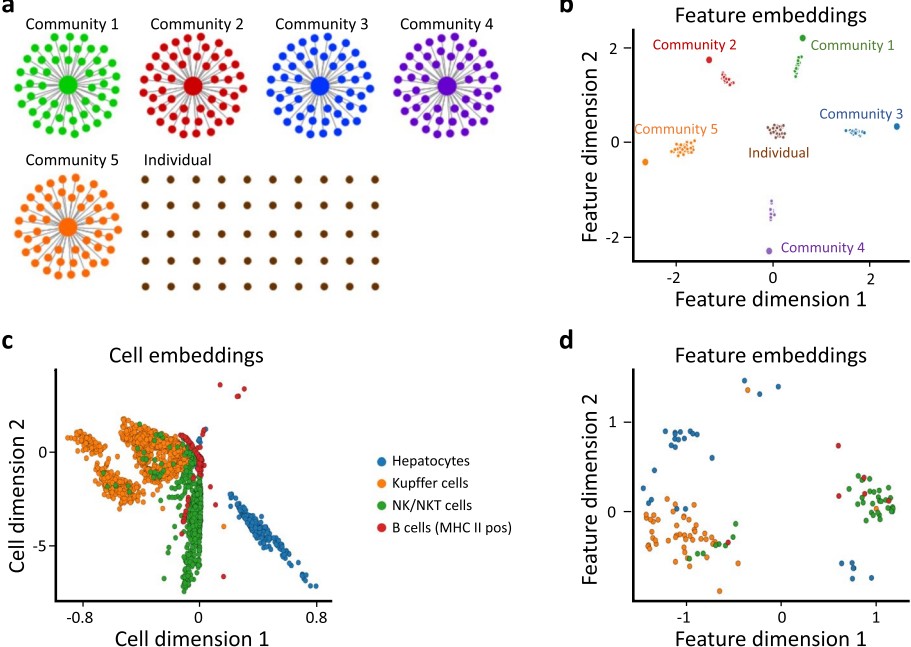

**Fig. 4** Co-expressed genes tend to co-localize in the feature embedding space. **a** The gene co-expression network used to simulate single-cell expression data for this experiment. The network consists of five tightly correlated groups of 50 genes each, along with 50 isolated, disconnected genes. Nodes represent genes, edges represent strong correlations. **b** Scatterplot of the feature embeddings inferred by siVAE trained on the dataset simulated from the network in **a**. Each point represents a gene, colored and labeled by the community it belongs to in **a**. **c** Scatterplot of the cell-embedding space inferred by siVAE trained on the fetal liver atlas dataset. Each point represents a cell and is colored based on its pre-defined cell type. **d** Scatterplot of feature embeddings inferred by siVAE trained on the fetal liver atlas dataset. Each point represents a marker gene and is colored based on its prior known association to a cell type

We found that genes belonging to the same community co-localized in the feature embedding space, but interestingly, the hub nodes are embedded in distinct locations their corresponding community (Fig. 4b). We reasoned that given the limited capacity of the cell-embedding space, siVAE tends to maintain information specifically about each hub because of their high degree centrality. On the other hand, non-hub genes within the same community co-localize in the feature embedding space because the limited capacity of siVAE forces non-hub genes to be predicted similarly, given the retained information about the hub. Interestingly, the 50 disconnected genes in the network were clustered tightly but near the origin in the feature embedding space, whereas genes that are part of a community are clustered but located farther away from the origin. This is likely because of two reasons. First, the KL divergence term of the feature-wise encoder-decoder of siVAE will tend to draw genes towards the origin. Second, because disconnected genes by definition do not co-vary with other genes, information about their expression pattern will tend to be lost during compression, leading the VAE to tend to predict the average expression level of that gene in the decoder (which will be 0 for all disconnected genes, because of data centering). This in turn encourages the feature embedding to be at the origin because the interpretability term encourages the linear product of the feature embedding with the feature loadings to predict the gene's expression pattern, so if a feature is located at the origin in the feature embedding space, it will cause the predicted expression to be 0. Additional file 1: Figure S11 confirms that genes close to the origin have higher reconstruction error. Note that when siVAE is trained without the interpretability term, the disconnected genes still tend to cluster but not necessarily near the origin (Additional file 1: Fig. S12).

We also confirmed that co-expressed genes cluster in the feature embedding space using the fetal liver cell atlas data. Unlike the simulations above, for the fetal liver atlas, there are no ground-truth GCNs available to use to identify truly co-expressed genes that are part of the same underlying gene communities. We therefore trained siVAE on the entire fetal liver atlas with 40 cell types and considered marker genes of the same cell type [60] to be a ground truth set of co-expressed genes. We selected four cell types from MSigDB for visualization (Fig. 4c), based on the observation that their established marker gene sets were least overlapping (suggesting they are distinct cell types) and that the marker gene sets are consistent with the CellTypist [61] database (Additional file 1: Fig. S13). For each selected cell type, we created a MSigDB meta-marker set by combining all gene sets available from MSigDB that corresponded to the target cell type. In the resulting feature embedding space learned by siVAE, we see that meta-markers of the same cell type tend to cluster in feature embedding space as expected (Fig. 4d). We also observed that visualizing markers from closely related cell types yields a more pronounced mixing of meta-markers from different cell types (Additional file 1: Fig. S14) because closely related cell types will tend to have a large overlap in meta-marker gene sets (Additional file 1: Fig. S13), and we would also expect their meta-marker genes are overall more closely co-expressed as well. Our results overall suggest that co-expressed genes tend to co-localize in the siVAE feature embedding space.

**Results – siVAE implicitly identifies gene hubs in the underlying co-expression network**

Our observation that hub genes in a community are treated differently by siVAE led us to hypothesize that we could identify hub genes from a trained siVAE model without inferring a GCN. Hub genes are often identified after GCN inference because they play essential roles both in terms of the structure of the network and the genome itself, and are often targets of genetic variants associated with disease [62, 63]. We reasoned that because hub genes are connected to many other genes, siVAE is more likely to store the expression patterns of hub genes in the cell embeddings for use in reconstructing both themselves and the rest of the gene expression patterns. We therefore used gene-specific reconstruction accuracy in the siVAE model as a GCN-free measure of degree centrality. As a ground truth measure of degree centrality, we calculated each gene's individual ability to predict the expression levels of every other gene in the genome (see Methods), reasoning that a 'hub' gene should be predictive of many other genes in the network. We observed a strong relationship between degree centrality and siVAE reconstruction accuracy as expected (Additional file 1: Fig. S15). Note for GCN-related analyses, we set the number of latent embeddings to 64 instead of 2 as was used for visualizing the embedding spaces, because here we do not need to directly visualize the embedding spaces.

Figure 5a compares the accuracy of siVAE's estimate of degree centrality against degree centrality estimated on GCNs inferred using a number of existing GCN inference algorithms (see Methods). Overall, siVAE has the highest correlation between its predicted degree centrality and the ground truth centrality (Spearman $\rho = 0.90$, $P = 2.2 \times 10^{-16}$), significantly larger than other approaches (median Spearman $\rho = 0.36$, $P = 9.0 \times 10^{-11}$). The strong correlation was also observed for siVAE with the interpretability term ($\gamma$) lowered to 0 (Additional file 1: Fig. S15). As a separate comparison of performance, we looked at the mean degree centrality (based on ground truth estimates) of the top 20 hubs, where larger values indicate stronger hubs. siVAE's mean degree centrality of the top 20 hubs was 0.092, larger than the GCN inference methods for whom the mean degree centrality of their top 20 predicted hubs is 0.074 (Fig. 5b). The results were also consistent when comparing the 2000 most highly variable genes (Additional file 1: Fig. S16). These results in total suggest that using siVAE, we can identify high-degree centrality genes more accurately than if we first inferred a GCN to then identify hub genes.

**Results – systematic differences in gene neighbors are observed between dimensionality reduction and network inference methods**

We also explored the extent to which we could identify neighboring genes that share an edge in a GCN, without having to infer GCNs explicitly. Gene neighbors tend to share similar function [64], interact with one another [65], and/or belong to the same gene community [66]. Identification of gene neighbors therefore helps identify co-functioning genes in the cell.

Here, we hypothesized that we could identify gene neighbors directly using a trained siVAE model, instead of having to first infer an explicit GCN. GCN inference methods typically output edge weights between pairs of nodes in the network, where larger

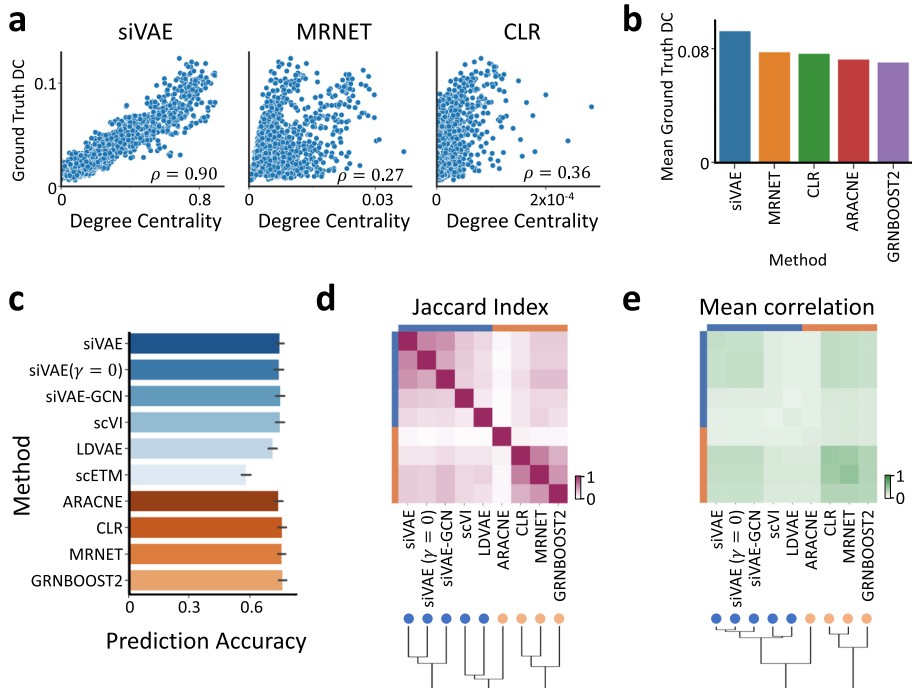

**Fig. 5** siVAE analysis yields insight into the underlying gene co-expression network structure of a cell population. **a** Scatterplot showing the correlation between ground truth degree centrality and predicted degree centrality based on either siVAE estimates or by computing node degree when a network is inferred using the MRNET or CLR algorithms. Each point represents a gene. **b** Average ground truth degree centrality of the top 50 genes ranked by predicted degree centrality across different methods. Higher average ground truth degree centrality indicates better concordance between the ground truth and predictions. **c** Bar plot indicating the prediction accuracy (% of variance explained) of the neighborhood gene sets when predicting each query gene, averaged over the 152 query genes with the highest predicted degree centrality in the fetal liver atlas dataset. Blue bars denote methods based on dimensionality reduction, while orange bars denote methods based on explicit gene regulatory network inference. **d** Heatmap indicating the pairwise Jaccard index (overlap) between neighborhood genes identified by pairs of methods. **e** Heatmap indicating the mean pairwise correlation in expression between neighborhood gene sets identified by pairs of methods

weights correspond to a greater chance the two nodes share an edge in the underlying GCN. For siVAE, we computed two different sets of edge weights: (1) siVAE-Euc, where the edge weight between two genes is set to their Euclidean distance in the feature embedding space, and smaller distances correspond to closer proximity; and (2) siVAE-GCN, where we first sample a new scRNA-seq dataset from a trained siVAE model that matches the size of the training data, then run a GCN inference method (ARACNE, MRNET, CLR, and GRNBOOST2) on the sampled scRNA-seq dataset to calculate edge weights between genes. To quantitatively evaluate the accuracy of neighbor identification using each method, we measured the percentage of variance explained of a given query gene when predicted by the expression levels of the nearest 20 genes ranked by edge weight to the query gene (see Methods). Intuitively, the true neighbors of a query gene should be more predictive of the query genes' expression levels compared to other genes. In our evaluations, we only consider the 152 query genes which were predicted to have high degree centrality across all tested methods (see Methods).

Overall, most methods identified neighbors that were equally predictive of the 152 query genes' expression levels, except for scETM (Fig. 5c). Excluding LDVAE and ARACNE, the

median percentage of variance explained for each method was 79.9% ± 0.84 s.d. Additional file 1: Fig. S17 illustrates that excluding LDVAE and ARACNE, the pairwise difference in percentage of variance explained between methods is only 0.013% on average. Notably, we observed a lower percentage of variance explained for LDVAE and ARACNE (on average, 77.2% variance explained, and 78.3% variance explained, respectively). The poorer results of LDVAE are consistent with its poorer classification performance results above.

When considering the overlap in neighbors selected by different methods, it is striking how the dimensionality reduction methods cluster strongly (scVI, siVAE, LDVAE) as a group, and the GCN inference-based methods cluster strongly as a group, but there is markedly less overlap between these two groups (Fig. 5d). This is surprising in part because the neighborhood sets are all approximately of the same predictive performance (Fig. 5c), suggesting the DR methods are systematically identifying different neighbors that are as equally co-expressed as the neighbor set identified by the GCN methods. In particular, consider that siVAE-GCN identifies gene neighbors using the GCN inference methodology, but is applied to a siVAE-generated dataset (instead of the original training dataset). Figure 5d and Additional file 1: Figure S18 illustrates that neighborhood genes identified by siVAE-GCN are still much more similar to those identified by siVAE than GCN inference methods, suggesting the unique neighborhood identified by the DR methods is a property of the co-expression patterns that DR methods learn, and not due to the method in which neighborhood genes are identified. The poor overlap between the DR and GCN methods also holds true if we consider the average pairwise correlation in expression between neighbor sets, instead of measuring direct overlap of genes (Fig. 5e). More specifically, the GCN-defined neighbor sets had higher average Pearson correlation among themselves (average Pearson $\rho = 0.67$, excluding ARACNE) compared to the average Pearson correlation coefficient among the neural net-based neighbor sets (average Pearson $\rho = 0.46$). There was also low average correlation between DR and GCN neighbor sets (average Pearson $\rho = 0.39$). Our results therefore suggest that since GCN- and dimensionality reduction-identified neighbor sets are systematically different but approximately equally predictive of neighboring genes, then both approaches should be used to find co-expressed genes in a network.

### Results – co-expression of mitochondrial genes in iPSCs are linked to neuron differentiation efficiency

We next hypothesized that we could use siVAE to implicitly learn GCNs across multiple cell populations, and in turn find associations between network structures and cell population-level phenotypes. Because robust single-GCN inference is already challenging, there has not been extensive work into approaches to comparing multiple GCNs [67–69]. As introduced earlier, the NeurDiff dataset includes scRNA-seq data collected across 215 iPSC cell lines profiled before differentiation (at 11 days, composed mainly of two progenitor cell types, mid brain floor plate progenitor (FPP) and proliferating FPP (P-FPP) cells), as well as after initiation of differentiation into neurons (day 30 and 52). By computing the fraction of sequenced cells at day 52 that were identified as mature neurons, each cell line has an estimate as to how efficiently they could be differentiated into neurons. Efficiencies were found to be highly reproducible, and the original study found promising associations between single gene expression levels of the pluripotent

cells at Day 11 and efficiency [37]. Here, we hypothesized instead that the co-expression patterns of a subset of genes in iPSCs at day 11 is also associated with differentiation efficiency (Fig. 6a). We performed GCN analysis on P-FPP and FPP cells separately.

We first selected a subset of 41 iPSC lines for analysis based on having sufficient numbers of cells (see Methods), then trained siVAE on each iPSC line to yield 41 gene embedding spaces. We then calculated a pairwise, implicit "GCN distance" between every pair of

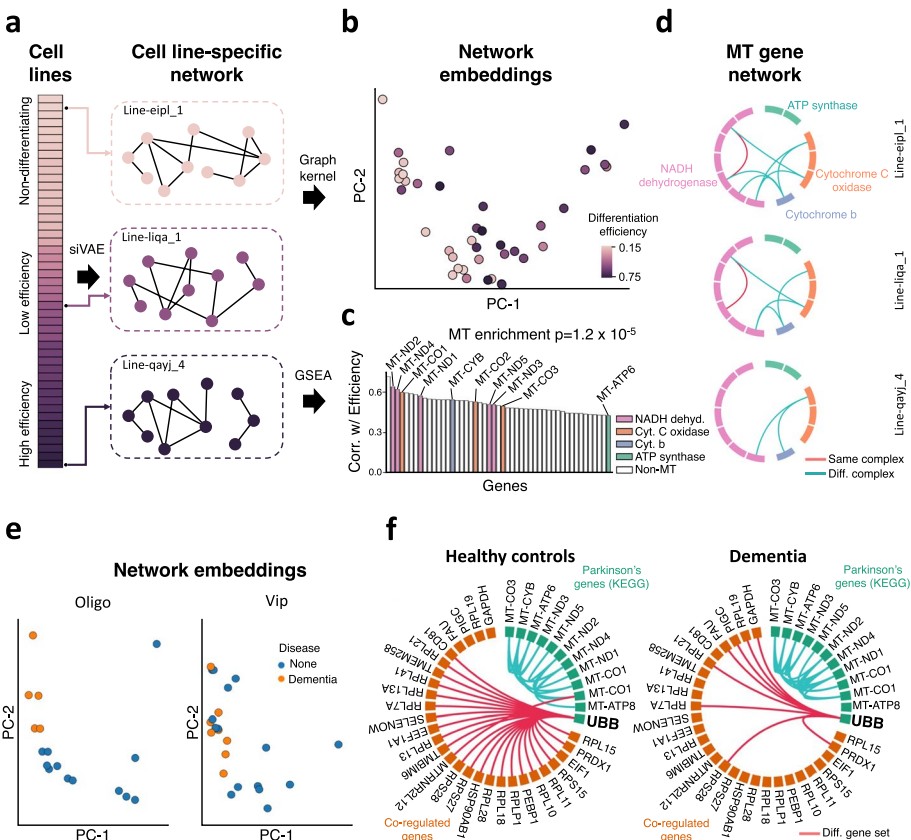

**Fig. 6** siVAE identifies gene modules whose connectivity correlates with cell population-level phenotypes. **a** Schematic illustrating how in experimental designs such as the NeurDiff dataset where there is a population of cells sequenced for each of several iPSC cell lines, then siVAE can be trained on each population separately to implicitly learn a gene co-expression network for each iPSC line. By conceptually ordering the cell lines by an iPSC line phenotype such as differentiation efficiency, siVAE can identify gene modules whose connectivity is correlated with differentiation efficiency. **b** Using a graph kernel, siVAE models representing different iPSC lines were embedded into a single scatterplot in which each cell line is represented by a single point, and two cell lines are proximal in the scatterplot if their respective gene co-expression networks are globally similar in structure. **c** Bar plot indicating the Spearman correlation between estimated degree centrality and differentiation efficiency for top correlated genes. Each bar indicates a gene, and the colored bars indicate mitochondrial genes. Each color represents a different mitochondrial complex. **d** Visualization of edges connecting mitochondrial genes in the siVAE inferred gene network for three selected cell lines. For each cell line, only seven edges between mitochondrial genes are considered for visualization; these seven edges represent those edges that are significantly negatively correlated with differentiation efficiency. **e** Scatterplot of donor embeddings in the DementiaDataset. Each point represents a donor's network, and the donor is colored by dementia case-control status. Network embeddings are shown for both oligodendrocytes and Vip interneurons. **f** Visualization of edges between the KEGG Parkinson's Disease gene set and their network neighbors, for both cases and controls. The edges were summarized for cases and controls separately by only keeping edges that were present in at least 20% of the donors within each case-control group

iPSC lines using a graph kernel on the gene embedding spaces (see Methods), then used the GCN distance matrix to embed the iPSC lines into a visualization using PCA (Fig. 6b). Surprisingly, we observed separation of cell lines according to their neuronal differentiation efficiency along PC-1, when visualizing all lines together (Fig. 6b) as well as when visualizing only iPSC lines that showed at least some differentiation (Additional file 1: Fig. S19). This result was unexpected because the graph kernel-based visualization (Fig. 6b) does not use any information about neuronal differentiation efficiency to guide visualization. The fact that the iPSC lines still separate by efficiency suggests that the major differences in GCN structure between iPSC lines are strongly associated with differentiation efficiency. We confirmed on both progenitor cell types (P-FPP and FPP) that differentiation efficiency explains separation in the cell line embeddings (Additional file 1: Fig. S19), but we focused on FPP for downstream analysis because of its larger dataset size relative to P-FPP.

To determine the genes whose varying GCN structure is most responsible for explaining variation between cell lines in Fig. 6b, we computed siVAE degree centrality for each gene and iPSC line and used gene set enrichment analysis (GSEA) to identify gene sets and pathways whose degree centrality correlated with differentiation efficiency. From the 125 genes whose degree centrality were significantly correlated with differentiation efficiency, we observed significant enrichment of mitochondrial (MT) genes (GSEA, adjusted $P=1.2\times10^{-5}$) (Fig. 6c). Individually, 10 mitochondrial genes had a high correlation between their degree centrality and efficiency (median Spearman $\rho=0.53$, $P=2.3\times10^{-5}$) (Additional file 1: Fig. S20).

A common explanation for change in connectivity for a single gene in a GCN is a change in average expression levels; genes that are turned off cannot covary with other genes, for example. To rule out this trivial explanation and focus on genes whose degree centrality is correlated with differentiation efficiency independent of changes in mean expression, we identified genes whose mean expression is correlated with differentiation efficiency. None of the 10 mitochondrial genes' mean expression levels were significantly correlated with differentiation efficiency (median Spearman $\rho=0.016$, median adjusted $P=0.75$, Additional file 1: Fig. S20). In contrast, 88 of the 115 non-MT genes whose degree centrality correlated with efficiency also demonstrated high correlation between mean gene expression levels and efficiency (median Spearman $\rho=0.21$, median adjusted $P=6.0\times10^{-6}$). The correlation between MT genes' degree centrality and differentiation efficiency is therefore not explained by changes in mean expression of MT genes.

Next, we examined the specific changes in the MT genes' network connectivity that were driving their correlation between degree centrality and efficiency. Based on the GCNs inferred from siVAE gene embeddings, the number of edges connecting one MT gene to one non-MT gene was consistently low across all cell lines with an average of 0.65 edges per cell line, and the number of such edges were not correlated with efficiency (Spearman $\rho=0.20$, $P=0.14$) (Additional file 1: Fig. S21). Connectivity between MT and non-MT genes therefore does not explain the variation in MT gene connectivity across lines. Instead, the correlation between MT degree centrality and differentiation efficiency is driven by changes in connectivity within the MT gene set (Spearman $\rho=-0.59$, $P=1.5\times10^{-6}$).

The mitochondrial genes encode for subunits of mitochondrial complexes. Given that our results above suggest that connectivity between MT genes is correlated with

differentiation efficiency, we sought to distinguish if it were connections between MT genes in the same or different complexes that are correlated with differentiation efficiency. We performed a Wilcox rank sum test on every pair of MT genes to test whether edges between or within MT complexes are correlated with differentiation. We identified seven MT edges correlated with differentiation efficiency in total, and of the seven correlated edges, six of them connect pairs of MT genes from different complexes. Of these six edges, three connect NADH dehydrogenase to cytochrome c oxidase, two edges connect NADH dehydrogenase to cytochrome b, and one edge connects cytochrome C oxidase to cytochrome b (Fig. 6d). This suggests that co-expression of distinct MT complexes is an important indicator of differentiation efficiency of these iPSC lines. Given the overall importance of MT genes and the potential role they play in differentiation efficiency, we then looked for genetic variants in MT genes that are associated with differentiation efficiency. Unfortunately, we were unable to identify genetic variants in the MT genome that were significantly correlated with differentiation efficiency (Additional file 1: Supplementary Note 2, Additional file 1: Fig. S22).

### Results – UBB loses co-expression patterns with multiple genes during dementia

To explore the extent to which associations between network structure and clinical phenotypes can be observed, we performed an analysis to identify network structures associated with dementia, using data from the Seattle Alzheimer's Disease Brain Cell Atlas dataset (DementiaDataset) [70]. We trained siVAE on each of five cell types (L2/3 neurons, L4 neurons, L5 neurons, oligodendrocytes, Vip neurons) from multiple donors to generate a gene embedding space per donor and cell type. We then used a graph kernel-based approach to visualize per-donor GCNs for the same cell types, and observed clustering of donors with respect to case-control status (dementia) for oligodendrocytes (Fig. 6e) ($P = 2.7 \times 10^{-6}$, Mann-Whitney $U$ test) and Vip interneurons ($P = 9.2 \times 10^{-3}$, Mann-Whitney $U$ test). These results suggest that dementia status is a strong driver of global variation in GCN structure. We focused on the oligodendrocyte cell type for the following analysis because of its larger number of cells. We computed siVAE degree centrality for each gene and for each donor and used GSEA to identify known gene sets whose degree centrality significantly correlates with dementia status. There was enrichment in one brain disorder-related gene set, the KEGG Parkinson's Disease set (GSEA, adjusted $P = 1.8 \times 10^{-7}$) that contains genes for which mutations are associated with Parkinson's Disease.

Given dementia is a common symptom of Parkinson's disease [71], we next performed network analysis to characterize how the network connectivity of these Parkinson's Disease genes varies with dementia status. Interestingly, while the connections between Parkinson's genes did not vary significantly with dementia status (average of 28 edges in healthy networks versus 29 edges in dementia networks), the number of edges between Parkinson's genes and other genes decreased from an average of 19 edges in healthy networks to 9 edges in dementia networks. Among the Parkinson's genes, UBB was the main contributor to this loss in edges, dropping from an average of 10 edges in donors with no dementia to an average of 3 edges in donors with dementia. Figure 6f visualizes the differences in network structure between cases and controls, where edges are only drawn if present in at least 20% of the case or control networks inferred by siVAE. We

see that the loss of connections of the Parkinson's genes is driven primarily by loss of connections by UBB to non-Parkinson's genes. These results highlight the potential for siVAE to help identify network structural features associated with clinical phenotypes (e.g., dementia status) that may indicate a role for changes in gene co-expression in disease etiology.

## Discussion

Through the development of siVAE, we have addressed one of the primary limitations of the interpretation of VAEs: the slow execution time of neural network feature attribution methods when the number of input features and embedding dimensions are both large. Single-cell atlases are ever-increasing in size due to the dropping cost of single-cell sequencing [72], thus yielding more complex collections of cells that warrant larger embedding dimensions when training VAEs. Also, there is rapidly increasing interest and development of multi-modal single-cell assays such as SNARE-Seq [73], ECCITE-Seq [74], and SHARE-Seq [75] that measure multiple data modalities (RNA, ATAC) simultaneously and are yielding single-cell measurements with up to hundreds of thousands of input features. Therefore, we expect the need to train VAE-based models with both larger embedding dimensions and larger numbers of input features will increase in the future, making approaches such as siVAE useful for generating interpretable dimensionality reductions.

Our analysis has also demonstrated how the interpretation of cell-embedding spaces can lead to insight into co-expression patterns of genes and identification of hubs genes underlying the cell population siVAE is trained on, without having to infer a GCN explicitly. This is useful because GCN inference continues to be a highly challenging task, even in the era of large numbers of cells sequenced from single-cell assays [76]. Furthermore, we showed how analysis across multiple cell populations with siVAE can identify co-expression patterns correlated with phenotypes of those cell populations, also without explicit GCN inference. The identification of network structures consistent with different phenotypes is not yet well explored in the literature, but potentially identifies phenotype-correlated changes in co-expression that cannot be detected by other means.

We made a surprising observation that the set of co-expressed neighbors of a given gene differs systematically depending on which approach (GCN inference, dimensionality reduction) was used to identify them. This is true even when a trained siVAE model was used to sample expression data that was then sent as input into a classic GCN inference method; in this scenario, the resulting siVAE-GCN yielded neighbors were still similar to those identified directly from siVAE. Our experiments further showed that both neighborhood sets are equally co-expressed with the query gene, suggesting at the least that accurate neighbor identification should leverage both GCN inference and DR methods. One possible explanation is that DR methods can learn to combine many genes into a single embedding dimension, whereas explicit GCN inference methods ultimately represent co-expression patterns as individual edges between only pairs of genes, and therefore are more limited in their capacity to represent higher order co-expression patterns.

Previous studies have established the importance of mitochondria in reprogramming, maintenance of pluripotency, and differentiation through their functional role in energy

production [77–79]. With respect to gene regulation, key mitochondrial transcription factors influence an iPSC's ability for differentiation [80–84]. The mean expression levels of established pluripotency markers such as SOX2, Oct4, Nanog, Klf4, and c-MYC [85] are correlated with differentiation efficiency [37]. Also, transcription factors associated with mitochondrial biogenesis (TFAM, POLG1, and POLG2) [79, 80] are necessary for successful differentiation. Our results showing co-expression of MT complexes as an indicator of differentiation efficiency are complementary in that there are few studies that have identified additional downstream genes [86, 87] associated with differentiation efficiency; prior work focused on identifying genes whose mean expression was correlated with differentiation efficiency, and did not identify MT genes [37].

There has also been recent work studying the impact of mitochondrial heteroplasmy on iPSC differentiation potential. Several studies now suggest mtDNA integrity as mandatory iPSC selection criteria [88–90]. Heteroplasmy of several mutations has been linked to an iPSC's ability to differentiate [89–92]. Manipulating MT heteroplasmy through insertion of wild-type mtDNA has been shown to revert diseased iPSC state and improve pluripotency [93]. Correlation of heteroplasmy with co-expression of MT complexes is an interesting avenue to pursue to determine whether heteroplasmy may be a cause of de-correlation of MT complexes.

While we have chosen the classic VAE framework upon which to build siVAE, our approach consisting of introducing a feature-wise decoder and interpretability term can be applied to other extensions of the VAE, such as VAE-GANs, $\beta$-VAE among others [94, 95]. With respect to genomic data modalities such as epigenomics, miRNA, and scRNA-seq, methods such as SCALE [96], RE-VAE [24], methCancer-gen [25], VAEMDA [27], scMVAE [26], scVI [20], Dr.VAE [32], scGen [33], and Dhaka [29] could also benefit from similar interpretability terms such as that used for siVAE. Many of these methods specifically focus on analysis beyond visualization [20, 28], such as trajectory inference [29, 97], data imputation [30], and perturbation response prediction [31–33]. An additional interpretability term could enable the identification of key input features in each task (e.g., identification of the set of regulatory genes tied to differentiation progression, the observed genes used to impute missing genes, and the genes affected by drug perturbation), which is a crucial step for validation and downstream application of these methods.

A promising application of siVAE is in multimodal data analysis. Assays such as SNARE-Seq [73] that jointly measure gene expression and chromatin accessibility from the same cell or CITE-Seq [98] that jointly measures gene expression and protein expression are useful for characterizing how variation in one modality (e.g., RNA) relate to variation in another modality (e.g., chromatin accessibility). For VAE-based multi-modal analysis methods (totalVI [99], multiVI [100], scMVP [101], BABEL [102], and Cobolt [103]), the siVAE interpretability term could be easily incorporated to identify mappings between features of different modalities, such as when linking enhancers to their target genes.

A related set of approaches to increasing the interpretability of generative models focuses on disentanglement learning. In particular, methods such as InfoGAN [104], FactorVAE [48], DirVAE [105], and others [95, 106, 107] modify generative models such as the VAE to achieve disentangled representations by encouraging the individual cell dimensions to be statistically independent. They show that independence between cell

dimensions oftentimes leads to more correspondence between individual cell dimensions and tangible factors such as width and rotation of digits for MNIST. However, we do not consider these model variants here because they do not provide contributions of individual features to cell dimensions. These approaches still require users to manually draw samples of points from the cell-embedding space, reconstruct the input features from the cell dimensions, then use human intervention to manually inspect how variation across specific dimensions might correspond to human-interpretable factors of variation. However, the regularization terms that encourages disentanglement between the cell dimensions may be applied to siVAE. This would help remove the entanglement between cell dimensions such as the overlapping outlines of digits in siVAE loadings for the MNIST dataset.

## Conclusions

Dimensionality reduction is a key first step in many single-cell genomic analysis tasks. siVAE represents a significant advance in making non-linear dimensionality reduction methods interpretable, thereby improving insight into factors that drive variation in the input data. We demonstrate how the interpretability features of siVAE enable identification of novel features of gene co-expression networks without requiring explicit network inference. These range from identification of gene hubs and gene neighbors to identifying gene modules whose network connections themselves are correlated with molecular and clinical phenotypes. siVAE is scalable and widely applicable beyond single-cell transcriptomics to other single-cell data modalities as well.

## Methods

### Model notation

We denote vectors as lower case, bold letters (e.g., $\boldsymbol{z}$). Matrices are upper case letters with two subscript indices (e.g., $X_{c,f}$). Constants are upper case letters with no subscripts (e.g., $L$).

### Generative process of VAEs

siVAE is an extension of a canonical variational autoencoder. Here we briefly review the generative process assumed by a canonical VAE with $L$ hidden layers in the decoder, and in which the hidden units of the last layer of the decoder are linearly transformed into the predicted mean of the Gaussian distribution over the observed data:

$$\boldsymbol{z}_{c,1} \sim N(0, I_K) \tag{1}$$

$$\boldsymbol{z}_{c,\ell} = \mu_\ell\left(\boldsymbol{z}_{c,\ell-1}\right), \ell = 2, \ldots, L \tag{2}$$

$$X_{c,f} \sim N\left(\boldsymbol{v}_f^T \boldsymbol{z}_{c,L}, \sigma_d\left(\boldsymbol{z}_{c,L}\right)\right) \tag{3}$$

$X_{c,f}$ is the input observed value for feature (e.g., gene) $f$ and cell $c$ (centered and scaled across all cells), where we assume there are $F$ features and $C$ cells in the training data. $\boldsymbol{z}_{c,1}$ is the embedding of cell $c$ in the (latent) cell-embedding space of the

VAE, while $z_{c,\ell}$ for $\ell > 1$ represent the activations of the hidden layer $\ell$ of the decoder for cell $c$. $v_f$ is the vector of incoming weights to the predicted mean of the output node $f$ of the VAE, while $\sigma_d(\cdot)$ is a one-layer function that predicts a non-negative scalar value representing variance. $I_K$ is the identity matrix of rank $K$. $\mu_1(\cdot)$, ..., $\mu_L(\cdot)$ represent the parameterized activation functions of hidden layers 1, ..., $L$ of the cell-wise decoder, respectively.

### Generative process of siVAE

The key idea behind siVAE is that we jointly infer cell-wise and feature-wise embedding spaces, and through regularization, loosely enforce correspondence between the cell and feature dimensions. Here, correspondence means variation in dimension $k$ in the cell-embedding space corresponds to observed variation in each feature $f$ that is proportional to feature $f$'s embedding coordinate in dimension $k$. Through correspondence, the feature embedding coordinates ("siVAE loadings") become analogous to PCA loadings, and the cell-embedding coordinates ("siVAE scores") become analogous to the PCA scores. In siVAE, the feature and cell embeddings are sampled from different latent spaces and projected to higher dimensions through separate decoders, before combining to produce the means of the Gaussians (Fig. 1a). The generative process assumed by siVAE is shown below:

$$z_{c,1} \sim N(0, I_K) \tag{4}$$

$$z_{c,\ell} = \mu_\ell(z_{c,\ell-1}), \ell = 2, \ldots, L \tag{5}$$

$$v_{f,1} \sim N(0, I_K) \tag{6}$$

$$v_{f,\ell} = \omega_\ell(v_{f,\ell-1}), \ell = 2, \ldots, L \tag{7}$$

$$X_{cf} \sim N\left(v_{f,L}^T z_{c,L}, \sigma_d(z_{c,L})\right) \tag{8}$$

Here $z_{c,L}$, $\mu_\ell(\cdot)$, $I_K$ and $\sigma(\cdot)$ are defined as above for VAEs. $v_{f,1}$ is the latent embedding of feature $f$ in the feature embedding space of siVAE, while $v_{f,\ell}$ for $\ell > 1$ represent the activations of hidden layer $\ell$ of the feature-wise decoder for feature $f$. $\omega_1(\cdot)$, ..., $\omega_L(\cdot)$ represent the activation functions of hidden layers 1, ..., $L$ of the feature-wise decoder, respectively. The schematic of siVAE neural network operations is outlined in Additional file 1: Figure S23.

Comparing Eqs. 3 to 6–8 illustrate that siVAE turns the last layer of weights leading to the Gaussian mean of the VAE into a non-linear transformation of the latent variables $v_{f,1}$. siVAE can therefore be viewed as putting a prior over a single (last) layer of weights in the VAE. The matrix $V = [v_{1,1}, \cdots, v_{F,1}]^T$ encodes the siVAE loadings, while the matrix $Z = [z_{1,1}, \cdots, z_{C,1}]$ encodes the siVAE scores. Note that we can

compute siVAE loadings and scores of other hidden layers $\ell$ as well, but in this paper, we focus on the latent space ($\ell = 1$).

### Inference and training

We employ variational inference via a pair of encoder networks, $\psi(X_{:,f})$ for features and $\phi(X_{c,:})$ for cells, in a manner analogous to variational inference applied to VAEs. Note the input for the two encoders is different: $X_{:,f}$ is a vector of observations for a single feature $f$ across all training cells, whereas $X_{c,:}$ is a vector of observations for a single-cell $c$ across all features. Our approximate posterior $q\left(\{v_{f,1}\}_{f=1}^{F}, \{z_{c,1}\}_{c=1}^{C}\right)$ factors as follows:

$$q\left(\{v_{f,1}\}_{f=1}^{F}, \{z_{c,1}\}_{c=1}^{C}\right) = \prod_{f=1}^{F} q(v_{f,1}) \prod_{c=1}^{C} q(z_{c,1}) \tag{9}$$

$$q(v_{f,1}) = N\left(v_{f,1}; W_{\psi}^{T} \psi(X_{:,f}), \sigma_{e,\psi}(\psi(X_{:,f}))\right) \tag{10}$$

$$q(z_{c,1}) = N\left(z_{c,1}; W_{\phi}^{T} \phi(X_{c,:}), \sigma_{e,\phi}(\phi(X_{c,:}))\right) \tag{11}$$

We perform variational inference and learning by maximizing the expected lower bound function $\ell_{\text{SIVAE}}$, where $\ell_{\text{KL}} = \text{KL}\left(q\left(\{v_{f,1}\}_{f=1}^{F}, \{z_{c,1}\}_{c=1}^{C}\right) \| p\left(\{v_{f,1}\}_{f=1}^{F}, \{z_{c,1}\}_{c=1}^{C}\right)\right)$.

$$\ell_{\text{SIVAE}} = -\ell_{\text{KL}} + \mathbb{E}_{q(z_{c,1}, v_{f,1})}\left[\sum_{c}\sum_{f} \log N\left(X_{c,f}; v_{f,L}^{T} z_{c,L}, \sigma_{d}(z_{c,L})\right)\right] \tag{12}$$

$$+ \gamma \mathbb{E}_{q(z_{c,1}, v_{f,1})}\left[\sum_{c}\sum_{f} \log N\left(X_{c,f}; v_{f,1}^{T} z_{c,1}, 1\right)\right] \tag{13}$$

### Interpretability term

The right-hand side of Eq. 12 is analogous to the KL divergence and reconstruction loss terms of the canonical VAE lower bound function. The term in Eq. 13, which we call the interpretability term, encourages the individual embedding dimensions of $v_{f,1}$ and $z_{c,1}$ to correspond to each other, by encouraging the linear product of $v_{f,1}$ and $z_{c,1}$ to approximate $X_{c,f}$. In our experiments, we set the penalty term $\gamma = 0.05$ to make the effect of the interpretability term small on the overall loss function.

### Reducing dimensionality of the input data for the feature-wise encoder-decoder

The size of input $X_{:,f}$ for the feature-wise encoder-decoder increases with $C$, the number of training cells. To avoid the computational expense of training with millions of cells (and therefore having an encoder with millions of input nodes), we reduce the

dimensionality of the input from $C$ to $C_{red}$ through either downsampling of cells or PCA. For downsampled input, we randomly sample $C_{red}$ cells while maintaining the ratio between the cell types. For PCA, we performed PCA without whitening on $X^T$, the input $G \times C$ data matrix, and retained the first $C_{red}$ principal components resulting in $X'^T$, a $G \times C_{red}$ score matrix. In Additional file 1: Figure S24, we show that training the feature-wise encoder-decoder with downsampled and PCA inputs results in loss function values and clustering accuracies comparable to that of siVAE trained with the full dataset.

### Training procedure for siVAE

We use a three-step training procedure to improve inference and learning:

- *Pre-train cell-wise encoder and decoder.* We first train the cell-wise encoder and decoder, similar to how a canonical VAE is trained, by optimizing the Eq. 12 component of $\ell_{SIVAE}$ with respect to $\{\mu_\ell, \sigma_d, \phi, \sigma_{e,\phi}, W_\phi\}$, and by treating the variables $\boldsymbol{v}_{f,L}$ as parameters to optimize to estimate $\widetilde{\boldsymbol{v}}_{(f,L)}$. The input to the cell-wise decoder are the cell-wise data points $X_{c,:}$ and the output are the same data points $X_{c,:}$.

- *Pre-train feature-wise encoder and decoder.* We next train the parameters associated with the feature-wise encoder and decoder, namely $\{\omega_\ell, \psi, \sigma_{e,\psi}, W_\psi\}$, by training a VAE whose inputs are the data features $X_{:,f}$, outputs are $\widetilde{\boldsymbol{v}}_{f,L}$ learned from the previous step, and whose encoder is defined by $\{\psi, \sigma_{e,\psi}, W_\psi\}$, and decoder parameterized by $\omega_\ell$, for $\ell = 1, ..., L-1$.

- *Train siVAE.* We finally train all model parameters $\{\mu_\ell, \sigma_d, \phi, \sigma_{e,\phi}, W_\phi, \omega_\ell, \psi, \sigma_{e,\psi}, W_\psi\}$ jointly by optimizing the full function $\ell_{SIVAE}$ from Eqs. 12 and 13.

### siVAE and VAE network design

A summary of the network designs used in this study can be found in Additional file 1:Table S1. For our experiments, identical neural network designs were used across the feature-wise and cell-wise encoders and decoders in siVAE. The architecture of the VAEs we compared against was matched to the architecture of the cell-wise encoder-decoders of siVAE. For MNIST and Fashion-MNIST, we set the architecture of the encoder to two hidden layers of sizes 512 and 128, and the decoder to two hidden layers of sizes 128 and 512. For all other datasets except the LargeBrainAtlas dataset, we set the architecture of the encoder to three hidden layers of sizes 1024, 512, and 128, and the decoder to three hidden layers of sizes 128, 512, and 1024. For LargeBrainDataset, we trained an encoder with three hidden layers of sizes 2048, 1024, and 512, and the decoder with three hidden layers of sizes 512, 1024, and 2048. We used a latent embedding layer with a size varying between 2, 5, 10, and 20 nodes for all imaging datasets for experiments testing the effect of a number of latent dimensions, and we used an embedding layer with a size of 2 for all other analyses. For the fetal liver atlas, we set the latent embedding layer size to be 2 for visualization tasks and for measuring clustering performance, and set the size to 64 for all other experiments (such as the GCN analysis). In the timing experiment, we varied the latent embedding layer size between 20, 128, and 512

for the LargeBrainAtlas dataset, while setting the latent embedding layer size at 2 for the BrainCortex dataset. For the NeurDiff dataset, we set the size of the latent embedding layer to be 32.

### siVAE and VAE model selection

We set model hyperparameters and optimization parameters by performing a hyperparameter search for the model with the lowest total loss on the held-out data. For each model, we used the Adam optimizer for training, with a learning rate of either 0.0001, 0.001, or 0.01. We considered L2 regularization with a scale factor $\lambda$ of either 0.001 or 0.01. For imaging datasets, we set the number of embedding dimensions to 20. For genomic datasets, we used two embedding dimensions for models that were used for visualization and clustering, and otherwise considered sizes of 16, 32, and 64 for all other analyses. All GCN-related analyses used 64 embedding dimensions.

### siVAE model variants

To explore the role of different design choices of siVAE, we created several variants of the siVAE model described above. siVAE ($\gamma = 0$) removes the interpretability term in Eq. 13 (by default, $\gamma = 0.05$). For comparison against LDVAE, whose decoder network ultimately predicts the parameters for negative binomial distributions, we also implemented both siVAE (NB) that predicts the parameters of a negative binomial distribution and VAE (linear) that is an identical implementation of LDVAE. For both siVAE variants that use the negative binomial distribution for modeling counts, raw counts were used as observations. siVAE (NB) is formulated as follows, where $l_\mu$, $l_\sigma$ parametrize the prior for scaling factor and are set to the empirical mean and variance of the observed data:

$$l_c \sim \log\text{normal}\left(l_\mu, l_\sigma^2\right) \tag{14}$$

$$\rho_{c,f} = \text{softmax}\left(v_{f,L}^{\text{T}} z_{c,L}\right) \tag{15}$$

$$m_{c,f} \sim \text{Gamma}\left(\rho_{c,f}, \sigma_d\left(z_{c,L}\right)\right) \tag{16}$$

$$X_{c,f} \sim \text{Poisson}\left(l_c m_{c,f}\right) \tag{17}$$

$$\ell_{SIVAE_{NB}} = -\ell_{\text{KL}} + \mathbb{E}_{q(z_{c,1}, v_{f,1})}\left[\sum_c \sum_f \log \text{Poisson}\left(X_{c,f}; l_c m_{c,f}\right)\right] \tag{18}$$

VAE (linear) is identical to siVAE (NB) except $v_{f,L}^T$ is replaced by $\phi_f$, an estimated parameter that matches the length of $z_{c,L}$, thereby removing the feature-wise encoder-decoder from the model. Finally, we implemented siVAE (linear), where the mean of the distribution over $X_{c,f}$ is directly predicted from linear multiplication of the cell

and feature embeddings. The reconstruction loss term corresponds to the interpretability term, eliminating the need for the latter.

$$X_{c,f} \sim N\left(\boldsymbol{v}_{f,1}^{T}\boldsymbol{z}_{c,1}, \sigma_d\left(\boldsymbol{z}_{c,1}\right)\right) \tag{19}$$

$$\ell_{\mathrm{SIVAE_{LINEAR}}} = -\ell_{\mathrm{KL}} + \mathbb{E}_{q\left(\boldsymbol{z}_{c,1}, \boldsymbol{v}_{f,1}\right)}\left[\sum_c \sum_f \log N\left(X_{c,f}; \boldsymbol{v}_{f,1}^{T}\boldsymbol{z}_{c,1}, \sigma_d\left(\boldsymbol{z}_{c,1}\right)\right)\right] \tag{20}$$

Batch correction is natively implemented in siVAE with a similar approach used in scVI [20]. $\boldsymbol{s}_c$ is a vector of length $b$ whose individual elements are either a continuous feature or a one-hot encoding of a categorical feature. The batch vector is concatenated to the input of the cell-wise encoder-decoder as well as the cell embedding to minimize the amount of batch effect captured in the cell embedding.

$$\boldsymbol{z}_{c,2} \sim \mu_2\left(\boldsymbol{z}_{c,1}\right) + \mu_s(\boldsymbol{s}_c) \tag{21}$$

Additionally, in the interpretability regularization term, we add weight $\boldsymbol{j}_f$, a vector of length $b$, that accounts for batch effect absence in the linear reconstruction in cell embedding.

$$\boldsymbol{\ell}_{\mathrm{SIVAE}} = -\ell_{\mathrm{KL}} + \mathbb{E}_{\boldsymbol{q}\left(\boldsymbol{z}_{c,1}, \boldsymbol{v}_{f,1}, \boldsymbol{s}_c\right)}\left[\sum_c \sum_f \log N\left(X_{c,f}; \boldsymbol{v}_{f,L}^{T}\boldsymbol{z}_{c,L}, \boldsymbol{\sigma}_d\left(\boldsymbol{z}_{c,L}\right)\right)\right] \tag{22}$$

$$+\gamma\,\mathbb{E}_{q\left(\boldsymbol{z}_{c,1}, \boldsymbol{v}_{f,1}\right)}\left[\sum_c \sum_f \log N\left(X_{c,f}; \boldsymbol{v}_{f,1}^{T}\boldsymbol{z}_{c,1} + \boldsymbol{j}_f^{T}\boldsymbol{s}_c, 1\right)\right] \tag{23}$$

### LDVAE and scVI

We used LDVAE [15] and scVI [20] implemented in the SCANPY [108] package available from PyPi. For model configurations of LDVAE and scVI that use the negative binomial distribution for modeling counts, raw counts were used as observations. The architecture of the model was set to match that of the cell-wise encoder-decoder of siVAE, including the number of dimensions of the cell-embedding space (set at two) and the number of hidden layers, as well as the number of hidden nodes. Model optimization was performed by varying the learning rate between 1e-2, 1e-3 and 1e-4, while the rest of the parameters were set to default. The models were trained for 100 epochs, and convergence of their loss functions during training was visually verified (Additional file 1: Fig. S25).

### scETM

We used scETM implemented in the package available from the original paper [16]. The architecture of the model was set to match that of the cell-wise encoder-decoder of siVAE, including the number of dimensions of the cell-embedding space (set at two) and the number of hidden layers, as well as the number of hidden nodes. Model optimization was performed by varying the parameters init_lr {5e-5, 5e-4, 5e-3}, max_kl_weight {1e-4, 1e-5, 1e-7}, and n_topics {50, 128, 256}, while the rest of the parameters were set to default. Per configuration, we trained the model for a varying number of epochs {50,100,500} and chose the trained state with the lowest loss. We show the result for the model with the optimal parameter.

### Feature attribution methods

Two separate Python packages were used to compute neural network feature attributions in our experiments. We used the DeepExplain Python package that implemented all feature attribution methods (Saliency Maps, Grad*Int, DeepLIFT, IntGrad, Shapley Value) included in our experiments in reverse-mode [109]. We used the tensorflow-forward-ad Python package for computing Saliency Maps and Grad*Int in forward-mode [110]. In both cases, the package applies feature attribution between the target nodes and input nodes. For application of feature attributions to the decoder, the target nodes and input nodes were set to be the nodes of the output layer and latent embedding layer, respectively, of the cell-wise decoder. For application of feature attributions to the encoder, the target nodes and input nodes were set to be the nodes of the latent embedding layer and input layer, respectively, of the cell-wise encoder. By default, the DeepExplain package summarizes the attribution across all target nodes, so binary masks corresponding to a single target node were used per target node. Similarly, the tensorflow-forward-ad package summarizes attribution across all input nodes, so binary masks corresponding to a single input node were used per input node. Integrated Gradients and DeepLIFT require an additional parameter of the input baseline, which represents a default null value that input values can be referenced against. We set this value to 0, representing the mean value of gene expression after preprocessing.

For Gene Relevance, we used the published R package [48]. The method required the latent embeddings learned from siVAE as well as the raw count data corresponding to the embeddings. We also varied the number of neighborhoods (10, 100, 1000, and default).

### Feature embeddings for feature attribution methods and Gene Relevance

All feature attribution methods tested here can output feature importance scores $s_{f,c}$ that represents a vector of contributions of feature $f$ to each embedding dimension for cell $c$. The Gene Relevance method [48] outputs partial derivatives in the same format. In contrast, siVAE loadings $v_{f,1}$ represents a vector of contributions of feature $f$ to each embedding dimension, summarized over all cells. To compare feature attribution methods to siVAE, we therefore need a procedure for converting the per-cell attributions $s_{f,c}$ into a set of overall feature attributions $u_f$ for each feature $f$ with respect to all embedding dimensions and that summarize across all cells, analogous to siVAE's loadings $v_{f,1}$. To do so, we

first construct a matrix $S_{d,f,c}$, containing all feature attributions for embedding dimension $d$, cell $c$, and feature $f$. For each embedding dimension $d$, we apply PCA to the 2D matrix $S_{d,:,:}$ to extract the first principal component's loadings $\boldsymbol{u}_{:,d}$, a vector of length $F$ that contains the contribution of each input feature $f$ to embedding dimension $d$. We repeated this process for each embedding dimension, then concatenated the resulting vector, resulting in the matrix $\boldsymbol{U}_{f,d}$, whose rows $\boldsymbol{U}_{f,:}$ are analogous to siVAE's $\boldsymbol{v}_{f,\,1}$. Finally, we calculated the Spearman correlation with a two-sided test between the feature embeddings inferred through different approaches per dimension and reported the median values.

### Generation of simulated scRNA-seq datasets from a gene network

To explore the organization of genes in siVAE feature embedding space, we simulated scRNA-seq data where the correlations between genes are consistent with a specified gene co-expression network. We designed a gene co-expression network that consisted of five communities of 50 genes each, as well as an additional set of 50 disconnected (isolated) genes that are independently varying. Each community included a single hub gene that was connected to the other 49 genes in the community, in a hub-and-spoke model. No other genes in the community were connected to any other gene. All edge weights representing pairwise correlations between genes in the same community were set to 0.6. The adjacency matrix capturing the co-expression patterns between the 300 genes were converted to a covariance matrix via the qpgraph R package [111], using the function qpG2Sigma with parameters rho=0.6. Afterwards, we used the resulting covariance matrix as input to a multivariate Gaussian distribution and sampled 5000 cells for training with siVAE.

### Cell type classification

The fivefold nested cross-validation experiments reported in Fig. 2b compare the performance of siVAE, VAE, and LDVAE on the fetal liver atlas dataset when matching their cell-wise encoder and decoder network designs. The number of embedding dimensions was fixed to be 2. After training using the training fold, the encoders were used to compute embeddings for the training and test datasets. We then used a $k$-NN ($k = 80$) classifier to predict labels of test cells based on the embeddings of the training and testing datasets. Similar five-fold nested cross-validation experiments were performed on the imaging datasets (MNIST, Fashion MNIST, CIFAR-10). However, we allowed the model to individually select the number of embedding dimensions $K$ from the set $\{2, 5, 10, 20\}$ using the training fold. In addition, the number of neighbors, $k$, was set to 15 as imaging datasets have far fewer samples than the fetal liver atlas dataset.

### Execution time experiments

We performed a series of experiments to compare siVAE training execution time against the combined execution time of VAE training and executing feature attribution methods. For Saliency Maps and Grad * Int, both forward and reverse modes were used. The majority of the feature attribution methods rely on taking the gradient of a single output node with respect to all input nodes using automatic differentiation in reverse-mode. For models with a large number of output nodes, the operation becomes computationally infeasible. Using automatic differentiation in forward-mode allows gradient calculation

of all output nodes with respect to a single input node, but faces the same computational issue for models with a large number of input nodes.

For the first experiment, we benchmarked using the LargeBrainAtlas dataset. In the case of the feature attribution execution times, we extrapolated the execution time on the LargeBrainAtlas dataset from the execution time on 100,000 cells, due to time constraints and the fact that runtime of these methods should scale linearly with the number of cells. In contrast, execution times of siVAE are based on the full dataset. For the second experiment, we tested the effect of varying either the number of embedding dimensions or the number of features on the execution time. As the execution time for two feature attribution methods (Integrated Gradients and Shapley Values) exceeded a realistic run time of 100 days, only the faster three methods (Saliency Maps, Grad * Int, and DeepLIFT) were used for the second experiment. For the LargeBrainAtlas dataset, the number of embedding dimensions was set to 20, 100, and 500. Similar to the first experiment, siVAE was run on the entire set of 1.3 million cells, and the VAE+feature attribution approaches were run on 100,000 cells and then linearly interpolated to the full dataset size. For the BrainCortex execution times, we varied the number of features by selecting the top $n$ highly variable genomic regions, where $n$ was set to either 28k, 120k, or 240k. We used a single NVIDIA GeForce GTX1080 Ti GPU, Intel Core i5-6600K CPU, and 32 GB RAM for all experiments.

### Estimating degree centrality using siVAE

We reasoned that the expression patterns of genes with high degree centrality are most likely to be retained by siVAE during dimensionality reduction, because those genes could be used to reconstruct the expression patterns of the many other genes connected to them. If so, then the hub genes are also likely to have the lowest reconstruction error. We therefore define degree centrality for siVAE as the negative reconstruction error of siVAE on each individual gene during training.

### Estimating degree centrality using GCN inference methods

The GCN inference methods tested here all output pairwise weights between genes, where larger weights indicated higher confidence in a pairwise edge in the underlying GCN. We therefore measured each query gene's degree centrality for GCN inference methods by averaging the weights between the query gene and every other gene in the network.

### Estimating the ground truth degree centrality

To compute the accuracy of siVAE-based degree centrality and GCN-based degree centrality, we generated ground truth degree centrality estimates as follows. We reasoned that a well-connected gene with high degree centrality would be highly co-expressed with many other genes in the genome. One way to quantitatively measure the degree of co-expression of a single query gene to all other genes is to measure how well the query gene can predict the expression level of all other genes in the genome. Therefore, our ground-truth degree centrality is defined as the percentage of variance explained by a query gene, with respect to all other genes in the genome. To measure percentage variance explained, for each gene in the genome, we trained a neural network consisting of a single input

node (corresponding to the query gene expression), 3 hidden layers with 128, 512, and 1024 nodes, and a final output layer of 2000 nodes for all remaining genes in the genome. The percentage of variance explained per gene was measured as $1 - \mathrm{Var}\left(X_{:,g} - \hat{X}_{:,g}\right)/\mathrm{Var}\left(X_{:,g}\right)$ where $\hat{X}_{:,g}$ is a vector of gene expression for gene g predicted across all cells by siVAE $X_{:,g}$. We then averaged the percentage of variance explained over all predicted genes and refer to this quantity as the ground truth degree centrality.

### Identifying gene co-expression network neighbors from siVAE models
GCN inference methods typically output a weighted adjacency matrix that indicates the strength of co-expression of every pair of genes, which in turn can be used to find the closest neighbors of every gene in the genome based on the largest values of the adjacency matrix. In our experiments in Fig. 5c, we used siVAE to also identify the closest co-expression neighbors of every gene in the genome, using two different approaches based on leveraging the feature embedding space. In our first approach (siVAE-Euc), we used Euclidean distance in the feature embedding space as a measure of distance between two genes in the network; the *k* nearest neighbors of a given query gene were defined as the *k* genes with the shortest distance to the query gene. Our second approach, termed siVAE-GCN, involved first sampling a new scRNA-seq dataset from a trained siVAE model that matches the size of the training data, then run a GCN inference method (ARACNE, MRNET, CLR, and GRNBOOST2) on the sampled scRNA-seq dataset to infer an explicit gene co-expression network. From this gene co-expression network, we extracted the nearest neighbors of every gene according to the strategy described below for GCN inference methods.

### Identifying gene co-expression network neighbors using GCN inference methods
For GCN inference methods, we defined a gene's local neighborhood as the closest 20 genes to each query gene based on largest pairwise weights for the query gene.

### Benchmarking gene neighborhoods
We characterized both siVAE and the GCN inference methods in terms of their ability to identify neighbor genes that are co-expressed. To do so, we applied each method to identify the 20 closest neighbor genes to each query gene. We then defined a prediction task in which the 20 neighbor genes were used as input to a neural network to predict the expression of the query gene. We used a fully connected neural network consisting of 3 hidden layers each with 16, 8, and 4 nodes, in addition to the input layer (with 20 nodes corresponding to the 20 closest neighbors), and the output layer consisting of a single node for the query gene. Accuracy was defined as the percentage of query gene expression variance explained. We compared siVAE to the GCN methods based on a set of 152 query genes, which were identified by computing the intersection of the top 500 highest centrality genes as determined by siVAE and each GCN inference method, to ensure that the query genes were unanimously considered of high degree centrality (and therefore should have many neighbor genes).

### Quantifying overlap in gene neighborhoods between siVAE and other methods

We also used two different strategies to gauge the overlap in the gene neighborhoods predicted by siVAE and each GCN inference method, defined as the 20 closest genes to every query gene. For percentage overlap, we measured the percentage of genes that overlapped between any two sets of neighborhood genes. For mean correlation, we measured the Pearson correlations of gene expression between every pair of genes between two neighborhood gene sets for the same query gene, then averaged the $20 \times 20 = 400$ correlation values together to compute mean correlation.

### Generating cell line embeddings of the NeurDiff dataset using siVAE and a graph kernel

We first divided the NeurDiff dataset into a single dataset per cell line. We performed downsampling of each iPSC line's data by splitting each cell line's collection of sequenced cells into equal-sized bins of 1000 cells, so that each siVAE model would be trained with the same number of cells. Remaining cells sequenced from a cell line that could not form a bin of 1000 cells were discarded. Each binned dataset was fed into siVAE for generating gene embeddings and siVAE-inferred degree centrality. We then inferred a GCN adjacency matrix from the gene embeddings, then used Weisfeiler-Lehman GraKeL [112] to generate a similarity matrix for each network. The similarity matrices generated from each of the bins corresponding to the same cell line were averaged together before PCA visualization of the cell lines.

### Isolating and visualizing changes in connectivity of the mitochondrial genes

For each gene, we computed the Spearman correlation between the gene's siVAE degree centrality and differentiation efficiency, in order to identify significantly associated genes using the Benjamini Hochberg-corrected P-values. For gene set enrichment analysis (GSEA) of the mitochondrial genes, we manually added an MT gene set consisting of all the genes starting with the prefix "MT-" to the curated set of KEGG pathways (http://www.gsea-msigdb.org/gsea/msigdb/collections.jsp) before performing GSEA using the prerank function from GSEAPY package on the Spearman correlation coefficients [113]. Spearman correlation between the mean expression of a gene and differentiation efficiency was measured through averaging the expression of a gene per line; *P*-values for Spearman correlation were also corrected using the Benjamini Hochberg procedure. Finally, we used the Wilcoxon Rank Sum test to detect which edges between mitochondrial genes were significantly associated with neuronal differentiation efficiency based on the inferred adjacency matrix. The mitochondrial gene network was visualized using Biocircos [114], showing only the significantly-associated mitochondrial edges.

### Testing for mitochondrial mutations associated with neuronal differentiation efficiency

Variant call files (VCF) for each iPSC cell line in the NeurDiff dataset were downloaded from the HipSci data browser (https://www.hipsci.org/lines/#/lines). We filtered for only mitochondrial variants by discarding all variants not found on the MT chromosome. We performed the Wilcox rank sum test on individual variants with respect to differentiation efficiency by treating the cell lines either with or without variants as two independent groups. Next, we performed gene-based burden testing where the variants were grouped into genes based on genomic position and jointly correlated with neuronal differentiation efficiency. In addition to grouping based on individual mitochondrial genes,

we also added an additional grouping for all mitochondrial variants. The Benjamini-Hochberg procedure was used to perform multiple testing correction.

### Generating donor embeddings of the DementiaDataset using siVAE and a graph kernel

We first divided the DementiaDataset into a single dataset per donor. So that each siVAE model would be trained with the same number of cells, we performed downsampling of each donor's data by splitting their collection of cells into equal-sized bins of 1000 cells. Remaining cells sequenced from a cell line that could not form a bin of 1000 cells were discarded. Each binned dataset was fed into siVAE for generating gene embeddings and siVAE-inferred degree centrality. We then inferred a GCN adjacency matrix from the gene embeddings, then used Weisfeiler-Lehman GraKeL [112] to generate a similarity matrix for each network. The similarity matrices generated from each of the bins corresponding to the same donor were averaged together before PCA visualization of the donor lines.

## Supplementary Information

---

**Additional file 1: Figures S1-S25, Tables S1-S3**, and **Supplementary Notes 1-2.**

**Additional file 2**. Review history.

---

**Peer review information**

**Review history**
The review history is available as Additional file 2.

**Authors' contributions**
GQ conceptualized and supervised the project. YC executed all of the computational analyses. RL aided in the design of the network analysis experiments. YC and GQ jointly wrote the manuscript. The authors read and approved the final manuscript.

**Funding**
This publication was made possible by an NIGMS-funded predoctoral fellowship to YC (T32 GM007377). GQ was supported by NSF CAREER award 1846559. This project has been made possible in part by grant number 2019-002429 from the Chan Zuckerberg Foundation. This work was supported by the National Institute of Child Health and Human Development P50 HD103526.

**Availability of data and materials**
*Code*
siVAE has been uploaded to the PyPi software repository and can be found here: https://pypi.org/project/siVAE/ https://github.com/quon-titative-biology/siVAE
A Zenodo repository of the software libraries is available here: https://zenodo.org/record/7495207#.Y68YuuzMl0Q [115]
All above repositories are available under the MIT license.
*Datasets*
A table summarizing the following datasets can be found in Additional file 1: Table S2.
*Fetal liver atlas dataset.* We obtained the fetal liver atlas [34] from ArrayExpress with accession code E-MTAB-7407 (https://www.ebi.ac.uk/arrayexpress/experiments/E-MTAB-7407/) on 2020/06/10, in processed form. We normalized the count matrix to TP10K (transcript per 10K transcripts), then performed feature selection by retaining the top 2000 highly variable genes, yielding 177,376 cells and 2000 genes. We then downsampled the number of cells to 100,000 cells, while preserving the proportion of cells from each cell type. Genes were individually centered and scaled to unit variance. For visualization of the feature embeddings in Fig. 4d, we obtained marker genes for four cell types (hepatocytes, Kupffer cells, NK/NKT cells, and MHC II positive B cells) that were available in the MSigDB database [113] (downloaded from http://www.gsea-msigdb.org/gsea/msigdb/collections.jsp#C8 on 2020/02/08). To account for the multiple subtype labels in the fetal liver atlas dataset matching to a single cell type in MSigDB, we allowed many-to-one mappings by grouping multiple cell type labels in the fetal liver atlas dataset that corresponded to one of the cell types in MSigDB. For each selected cell type, we combined all the MSigDB gene sets corresponding to the target cell type to create a MSigDB

meta-marker set. The exact groupings are shown in Additional file 1: Table S3. We only visualized the cells with known marker genes, and genes that belonged to more than one marker gene set (shared across cell types) were discarded.
*MNIST and Fashion-MNIST dataset.* We obtained both datasets from the TensorFlow datasets web page on 2020/02/20. Images were flattened, centered, and scaled to unit variance per feature across all images before input into the models.
*CIFAR-10 dataset.* We obtained CIFAR-10 from the TensorFlow datasets web page on 2020/02/20. We then subsampled the image classes to only the airplane and ship classes because other image classes require convolutional layers to achieve reasonable classification performance, and were thus unsuitable for benchmarking VAEs. Images were flattened, centered, and scaled to unit variance per feature across all images before input into the models. Color channels were concatenated and flattened.
*LargeBrainAtlas dataset.* We obtained the 1.3 Million Brain Cells dataset referred to as "LargeBrainAtlas" from the 10x Genomics website (https://support.10xgenomics.com/single-cell-gene-expression/software/pipelines/latest/advanced/h5_matrices) on 2020/04/28. We normalized the count matrix to TP10K, then retained all genes. After, genes were individually centered and scaled to unit variance.
*BrainCortex dataset.* We obtained the BrainCortex dataset (GSE126074) from (https://www.ncbi.nlm.nih.gov/geo/query/acc.cgi?acc=GSE126074) on 2020/12/01. We performed quality control based on TSS enrichment and nucleosome signal which filtered the dataset down to 244,544 features.
*NeurDiff dataset.* We obtained the iPSC neuronal differentiation dataset referred to as the "NeurDiff" dataset from https://zenodo.org/record/4333872 accessed on 2021/12/10. We only used the gene expression matrix for day 11 (D11) with pre-normalized expression. We divided the dataset according to cell type (FPP and P-FPP), then filtered out the cell lines that contained less than 1000 cells of that cell type. The downstream preprocessing and experiments were performed per cell type. For each cell type individually, batch correction was performed per cell line to regress out pool_id, as well as cell cycle score using the regress_out() function from scanpy [108]. Next, we performed feature selection by taking the union of the top 2000 genes with highest variance in each cell line as well as the top 2000 genes with highest variance across all cell lines. Afterwards, genes were individually centered and scaled to unit variance. The final dataset consisted of 41 cell lines (109,483 cells) and 3362 genes for P-FPP, and 27 cell lines (85,961 cells) and 3308 genes for FPP.
*DementiaDataset.* We obtained the SEA-AD: Seattle Alzheimer's Disease Brain Cell Atlas dataset from https://cellxgene.cziscience.com/collections/1ca90a2d-2943-483d-b678-b809bf464c30 accessed on 2022/09/25. We selected 5 cell types (L2/3 neuron, L4 neuron, L5 neuron, oligodendrocyte, Vip) with the largest number of cells, with a minimum of 100k cells. Per cell type, we then filtered out the donors that contained less than 1000 cells of that cell type. The downstream preprocessing and experiments were performed per cell type individually. For each cell type, batch correction was performed per donor to regress out pool_id as well as cell cycle score using regress_out() function from scanpy [108]. Next, we performed feature selection by taking the union of the top 2000 genes with the highest variance in each donor as well as the top 2000 genes with the highest variance across all donors. Afterwards, genes were individually centered and scaled to unit variance. The final dataset consisted of 58 donors (308,874 cells) and 3384 genes for L2/3_IT, 36 donors (99,060 cells) and 4863 genes for L4_IT, 13 donors (39,894 cells) and 3449 genes for L5_IT, 21 donors (44,327 cells) and 3141 genes for Oligo, and 24 donors (41,785 cells) and 4438 genes for Vip.

*Databases*
Fetal liver atlas dataset.
(https://www.ebi.ac.uk/arrayexpress/experiments/E-MTAB-7407/) [34]
Imaging datasets (MNIST, Fashion MNIST, CIFAR-10).
(https://www.tensorflow.org/datasets/catalog/) [116–118]
LargeBrainAtlas dataset.
(https://support.10xgenomics.com/single-cell-gene-expression/software/pipelines/latest/advanced/h5_matrices) [50]
BrainCortex dataset.
(https://www.ncbi.nlm.nih.gov/geo/query/acc.cgi?acc=GSE126074) [51]
NeurDiff dataset.
(https://zenodo.org/record/4333872) [37]
DementiaDataset.
(https://cellxgene.cziscience.com/collections/1ca90a2d-2943-483d-b678-b809bf464c30 accessed on 2022/09/25) [70]

## Declarations

### Ethics approval and consent to participate
Ethics approval is not applicable.

### Competing interests
The authors declare that they have no competing interests.

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

## 
