## [**Additional file 2**. Review history. · Genome Biology]

Review History

First round of review

Reviewer 1

Were you able to assess all statistics in the manuscript, including the appropriateness of statistical tests used? No.

Were you able to directly test the methods? No.

Comments to author:

I think this approach is really creative and useful! This approach seems so obvious retrospectively, but you are the first to do it, so congratulations on this rigorous and creative method! I think the paper is great, there are only a few points that I need to mention. In summary, I think everything checks out, but there were some aspects of the methods that needed a bit more clarification for me to fully understand. The biggest points to me were:

- 1) kNN classification accuracy as an evaluation metric. Since the cell-types vary greatly in number how do you know the accuracy isn't biased by one large cell type
- 2) Better justification for using so few cell types in figure 4c/d, and also using non-canonical cell-type markers. To me, at least, I don't reach for MSigDB for cell-type markers. I put a link to an alternative below. Furthermore, one could at least take the top differential genes between cell-types as marker genes.
- 3) More details in the methods. There are several points I mention below.
- 4) Some findings seem to be supported via visual inspection, but I think some easy statistics could be used to strengthen the authors points. Specifically figure 4d, 6b, S15

Code:

Currently your code requires tensorflow-1.15 for installation, but the README states ≥ 1.15 . I wasn't able to install it due to this requirement. Downgrading to such a low version of tensorflow is painful. Requirement shown here: <https://github.com/quon-titative-biology/siVAE/blob/b1883e6e245966652a66c7d6d0d978f2f1e7eeb3/setup.py#L61>

However, in regards to the other code it looks like it could be used to recreate all the figures and analyses pretty easily. I also really appreciate the tutorial!

Text:

- Using the 5-fold cross-validation framework -- what does accuracy mean? Is this balanced accuracy? Given that the number of cells of each cell type is not the same, how do you know accuracy isn't biased by a large cell type? I would suggest reporting per-class precision/recall as well.

- Page 7, there is a reference to Supplementary Fig. S6, do you mean S7?

- Page 10, line 9, you state earlier that there are 41 cell types, but here state there is only 40 cell types.

- I think there needs to be more justification as to why only four cell types are used in figure 4c/d. There exist more annotations of marker genes for cell types. Currently, CellTypist aggregated multiple markers across several datasets, including the dataset of interest, Popescu et al. 2019. You can find all the literature based markers aggregated by CellTypist here: https://github.com/Teichlab/celltypist_wiki/blob/main/atlasses/Pan_Immune_CellTypist/v1/encyclopedia/encyclopedia_table.xlsx. While the markers are fewer, it spans more cell-types. They may also be more cell-type specific than the markers you are currently using which may result in a cleaner Fig. 4d. Currently in Fig. 4d, I would argue that markers of **some** cell types tend to cluster together.

- In the methods section titled "siVAE and VAE network design" it states: "Additional implementation details, as well as a table containing the above information on network design, can be found in the Supplementary Materials (Supplementary Table 1)." I wasn't able to find the additional implementation details, could you include a more specific pointer to where this is?

- In Supplementary Table 1, I am unsure which latent dimensions were used for which experiment. This is clear for the fetal liver atlas, 2D for visualization and 64D for all other analyses. But what about the other datasets? Were these all used for a different experiment or only considered in the hyperparameter search for a good latent dimension?

- In addition, it's not readily apparent to a casual reader that in some cases two models are being trained - one for visualization and one for other downstream tasks. I think this should be mentioned in the main text and not only in the methods section.

- In the methods section titled "Cell type classification." it states: "In addition, the number of clusters, k , was set to 15 as imaging datasets have far fewer classes than the fetal liver atlas dataset." I am confused by the notation here, isn't k the number of neighbors used, not the clusters?

- Similarly, the same section states: "After training using the training fold, the encoders were used to compute embeddings for the training and test datasets. We then used a k -NN ($k = 80$) classifier to predict labels of test cells based on the embeddings of the training and testing datasets." I'm just a bit confused on the wording and want to make sure I got it correctly. So, after cross-validation was performed, you used your training set to get your final trained embedding. Using this trained embedding, you applied it to the combined training and test data sets. Then you learned a single kNN classifier on the embedded training data and evaluated the trained kNN model on the test data. Is this correct?

- Just to make sure I am understanding, for the fetal liver atlas, was the kNN classification done on the 64-dimensional latent embedding?

- In the method section titled "Execution time comparison experiments." It states "In the case of the feature attribution execution times, we extrapolated the execution time on the LargeBrainAtlas dataset from the execution time on 100,000 cells, due to time constraints and the fact that runtime of these methods should scale linearly with the number of cells. In contrast, execution times of siVAE are those measured on the full dataset." I think this is potentially misleading. I am naive, so I do not know if these "methods should scale linearly". I think this needs to be shown in some capacity, or at a minimum clearly stated in the figure caption.

- In the method section titled "Generating cell line embeddings with siVAE and graph kernel." it states "We then used in Weisfeiler-Lehman GraKeL111 to infer adjacency matrix per gene embedding then generate similarity matrix between inferred network generated per binned datasets." I would recommend rewording it a bit for people unfamiliar with GraKeL, it sounds like GraKeL only infers the adjacency matrix, then you generate a similarity matrix from it. But I think you mean to say that you generate the adjacency matrix from the gene embeddings then use GraKeL to generate the similarity matrix.

- Page 12, line 24 it states "Surprisingly, we observed separation of cell lines according to their neuronal differentiation efficiency along PC-1, when visualizing all lines together (Fig. 6b), as well as when visualizing only iPSC lines that showed any differentiation (Supplementary Fig. 12)." Can you calculate a correlation statistic here? PC-1 vs differentiation efficiency? Also I believe Supp Fig 12 should be 15.

- Page 12, line 32 it states "We confirmed on both the two progenitor cell types (P_FPP and FPP) that differentiation efficiency explains separation in the cell line embeddings (Supplementary Fig. 15)," Is this from visual inspection? I am not sure I see this same separation -- can a correlation statistic be included here? I am most confused by the column "Diff. Cells Only", I don't see a clear separation, but maybe I am interpreting things incorrectly? Also, S15 is missing a color key.

- Line 49 page 6 it says "We tested our model on an iPSC neuronal differentiation (NeurDiff) dataset in which XX cell iPSC..." what is XX?

- TP10K is used twice in the methods but is not explained. It would be nice to write it out once for readers unfamiliar with this acronym.

Reviewer 2

Were you able to assess all statistics in the manuscript, including the appropriateness of statistical tests used? Yes.

Were you able to directly test the methods? No.

Comments to author:

The authors proposed a VAE model called siVAE for single-cell RNA-seq data. The main novelty of the model is the use of two encoders instead of one encoder as in the standard VAE for encoding both cells and genes. The decoder then takes the inner product of the cell embedding and feature embedding to produce the reconstructed cell expression. To encourage interpretability, additional data likelihood on the embedding generated from the first layer of the cell encoder and feature encoder was added to the

standard ELBO. While the overall methodological innovation is modest, the authors advocated the interpretability side of the method especially on predicting gene co-expression network and have done a great amount of work on experimenting model variants and biological interpretation.

Major comments:

1. The proposed method has a lot of resemblance of scETM (Zhao et al., Nat Comm 2021) (source code: <https://github.com/hui2000ji/scETM>) except for a few technical differences. Although authors cited the paper, they did not perform thorough comparison with scETM as they did with LDVAE. I suggest that they compare siVAE with scETM in both cell type classification (Figure 2) and query gene prediction (Figure 5).
2. The feature encoder size grows as the number of training cells, which is a main issue when dealing with millions of cells (e.g., LargeBrainAtlas). The authors remedied this issue by either downsampling cells or applying PCA to get C_red eigenvector PC cells. However, it is not clear to me how the feature encoder applies to test cells. My guess is the feature encoder is discarded and only the cell encoder is used to generate embedding for a test cell, which is the same as the standard VAEs. Therefore, the feature encoder is not transferable. Please verify.
3. Since the feature embedding and cell embedding are both unbounded real value with no L1 sparsity regularization or non-negative constraint, interpreting feature embedding based on whether it's highly positive (i.e., as illustrated in Figure 1d) can be misleading. The problem is that a highly positive values in the score embedding space can be canceled out by the high negative values in the loading embedding space when performing the reconstruction so the net result is the same as having zero score and loading. This is the reason why non-negative matrix factorization or topic models are usually preferred choices to interpret learned latent factors over PCA or probabilistic PCA. Can authors comment on that?
4. The authors used KNN for cell type classification. Adjusted Rand Index (ARI) is also a commonly used metric especially for evaluating unsupervised clustering methods where the cell types are the groundtruth cluster labels and the Louvain or Leiden cluster indices using the model-dependent embedding are the predicted clusters. Please consider adding ARI as another metric when comparing with the existing methods.
5. Authors experimented Negative Binomial as the data likelihood siVAE (NB) as opposed to Gaussian likelihood used in siVAE and saw worse performance. Can authors clarify whether their siVAE (NB) operates on log-transformed data or raw count data?
6. The authors defined hub genes as the genes with high reconstruction accuracy compared to other genes. However, house keeping genes may also be well predicted since their expression don't change much but they are often not hub genes. Can author explain that?
7. Related to comment 6, are the identified hub genes TF master regulators? Can authors show that the gene co-expression network inferred by siVAE recovers any known regulatory or interaction network information such as TFBS, PPI, chromatin co-accessibility, etc?
8. The simulation presented in Figure 4a,b and described in the main text is a little too simplistic. It's a toy data for sanity check. Author should consider moving it to supplementary (as they did for MNIST).

Methods description:

9. The description for the generative processes of both VAE and siVAE are not accurate. Starting with the standard VAE, the generative process does not involve any neural network, i.e., $z_c \sim N(0, I_K)$; $X_c \sim N(\theta z_c, \sigma I)$. It's the variational inference that involves it i.e., $q(z_c|X_c) \sim N(\mu(X_c), \sigma(X_c))$. Similarly, the generative process of the siVAE should be: (a) $z_c \sim N(0, I_K)$; (b) $v_f \sim N(0, I_K)$; (c) $X_{c,f} \sim N(z_c v_f, \sigma)$. It's the variational inference involves network encoders $q(z_c|X_c)$ and $q(v_f|X_f)$. Having a neural network in the generative process itself will lead to non-closed-form ELBO since the prior distribution of the network-transformed Gaussian (i.e., $f(z_c, l)$ for $l > 1$) is not a Gaussian.
10. The authors mentioned that they focused on latent space of the first layer of both cell encoder (i.e., siVAE scores) and feature encoder ($l=1$) (i.e., siVAE loadings), which means the purpose of the deep

layers $l > 1$ are mainly for better reconstruction. However, in standard VAE, it's the output from the last layer L (i.e., $[z_{1,L}, \dots, z_{C,L}]$ and $[v_{1,L}, \dots, v_{F,L}]$ in the context of siVAE) that are used for downstream analysis. The rationale of using the first layer $l=1$ is unclear.

11. In Equation 12 for the ELBO (which in its correct form should be: $ELBO = E[\log p(X|z) + \log p(z) - \log q(z)] = E[\log p(X|z)] - E[\log q(z) - \log p(z)] = E[\log p(X|z)] - KL[q(z)||p(z)]$, the minus sign is missing in front of l_KL (i.e., $E[\log q - \log p]$). Same error in Equation 18 and 20. Also, equations 13 and 23 (i.e., the interpretability term) should have a plus sign instead of minus sign (since you want to encourage linear reconstruction for interpretation). I am not sure whether this is a conceptual understanding mistake or the authors in fact implemented the wrong objective functions the same way as they described.

Minor comments:

1. I suggest avoid using GCN as its connotation is Graph Convolutional Network in the deep learning community. I personally found it quite confusing since GCN has been used throughout the Methods section that I started reading before the Results section. Maybe GCEN is a better abbreviation.
2. The paper needs careful proofreading. Page 22 line 35, "The five-fold nested cross validation experiments reported in Figure 1c", I think "Figure 1c" is the wrong reference
3. Figure 1 legend "Note the embeddings of genes 1, 2, 3 and 4 all have large magnitudes along dimension 1 but not dimension 2," should be describing panel d not panel b.
4. Figure 2 "embeddfing" => embedding

Reviewer 3

Were you able to assess all statistics in the manuscript, including the appropriateness of statistical tests used? Yes.

Were you able to directly test the methods? No.

Comments to author:

In the work "Interpretable deep generative models for genomics" Choi and coworkers present a new technique for generating interpretable variational auto-encoders, siVAE. They test their methods on siRNA-seq data-sets and show that it works well in clustering known cell-types together and also that gene-gene associations from siVAE can be interpreted as a co-expression network where the topology quite well corresponds to alternative network inference approaches and lastly shows how hubs can be used for associating genes to neuronal differentiation. In general I find the technique very timely and important for the field and indeed this tool could help many researchers in the si field. So I am very positive to the work presented here. I have a few comments below:

1.) In the title they use genomics, but it is not tested on genomic associations, which is misleading. Better to use siRNA-seq in title as well.
2. Authors spend substantial time comparing it to other methods, which is good, However they don't show so much external biological validations. An exception is the pathway associations and neuronal differentiation, which I liked. However, the role of hubs are key in complex diseases and cancers. Given the great variety of readers in Genome Biol I believe a si data-sets and topological analysis in such a data-set would be showing how the method could be used for

precisions medicine applications.

3. It would be interesting if authors discussed how one could take this to another level using multi-omics siRNA-seq data. It feels like this could be very good data for this model associating for example ATAC-seq and RNA-seq. This could be done in a forthcoming paper.

4. A minor comment is that figure 2 says that $\gamma=0.05$ was used, but other places indicate $\gamma=0$. Seems to be a typo there.

We thank the reviewers for their helpful feedback and constructive criticism. Below, we address their concerns:

Reviewer #1

Major comments

1. kNN classification accuracy as an evaluation metric. Since the cell-types vary greatly in number how do you know the accuracy isn't biased by one large cell type.

We now compare performance using metrics including cell type-balanced classification accuracy and ARI. The updated results in (**Fig. S2,S3**) shows that siVAE still outperforms its competitors across different metrics.

2. Better justification for using so few cell types in figure 4c/d, and also using non-canonical cell-type markers. To me, at least, I don't reach for MSigDB for cell-type markers. I put a link to an alternative below. Furthermore, one could at least take the top differential genes between cell-types as marker genes.

We agree that markers for individual cell types at lower levels of the cell type hierarchy (where there would then be more cell types) is desirable. However, as we move lower in the hierarchy, the cell types themselves generally become harder to distinguish (both at the genome-wide expression profile level, and their corresponding markers).

We have tried using the suggested CellTypist marker genes to annotate the gene embeddings (see the related minor comment below for details) and found that as expected, clustering of marker genes at lower hierarchy of cell type was more difficult (**Fig. S13b,d**) compared to the higher hierarchy (**Fig. S13a,c**). We further observed that there is a high degree of overlap between marker genes of different cell types across both the higher and lower hierarchy annotations of CellTypist (**Fig. S14**), further demonstrating that these cell types are very closely related.

We have now added more justification to the text.

3. More details in the methods. There are several points I mention below.

We have clarified the details suggested in the minor comments below.

4. Some findings seem to be supported via visual inspection, but I think some easy statistics could be used to strengthen the authors points. Specifically figure 4d, 6b, S15

We have added the suggested statistics into the main and supplementary text as discussed below in the minor comments.

Minor comments

1. Using the 5-fold cross-validation framework -- what does accuracy mean? Is this balanced accuracy? Given that the number of cells of each cell type is not the same, how do you know accuracy isn't biased by a large cell type? I would suggest reporting per-class precision/recall as well.

The classification accuracy is calculated as (# of correctly classified cells / # of total cells). As pointed out by the reviewer, the previous accuracies were not balanced. To address this, we have calculated the class-balanced accuracy as well as ARI to show that these results are consistent with our original accuracies with siVAE outperforming other methods beside VAE and the ranking of the performances staying mostly consistent as seen in (Fig. S2).

We replaced the original classification accuracy bar plot in **Figure 2b** with the balanced accuracy.

In addition, we plotted the per-class accuracies as suggested, in order of the largest to smallest cell types (Fig. S3 shown below). We observed that the per-class accuracies for cell types with smaller number do tend to have slightly lower accuracies, but the difference is not large for reasonably-represented cell types (# of cells > 100). In addition, this pattern occurs across all methods. siVAE still outperforms other methods (with the exception of the VAE) in the majority of the cell types.

We have updated the result to main text: “Furthermore, siVAE is competitive in (balanced) classification accuracy with a classic VAE on the fetal liver cell atlas both across all cell types (**Fig. 2b**, **Fig. S2**) as well for each cell type individually (**Fig. S3**).”

2. Page 7, there is a reference to Fig. S6, do you mean S7?

We have fixed the typo.

3. Page 10, line 9, you state earlier that there are 41 cell types, but here state there is only 40 cell types.

For the FetalLiver dataset, there should be 40 cell types, and has been fixed in the paper: “We benchmarked siVAE against other interpretable and non-interpretable dimensionality reduction approaches using a fetal liver cell atlas³⁴ consisting of 177,376 cells covering 40 cell types.”

4. I think there needs to be more justification as to why only four cell types are used in figure 4c/d. There exist more annotations of marker genes for cell types. Currently, CellTypist aggregated multiple markers across several datasets, including the dataset of interest, Popescu et al. 2019. You can find all the literature based markers aggregated by CellTypist here: https://github.com/Teichlab/celltypist/wiki/blob/main/atlas/Pan_Immune_CellTypist/v1/encyclopedia/encyclopedia_table.xlsx. While the markers are fewer, it spans more cell-types. They may

also be more cell-type specific than the markers you are currently using which may results in a cleaner Fig. 4d. Currently in Fig. 4d, I would argue that markers of **some** cell types tend to cluster together.

While we agree that markers for individual cell types at the lower levels of the cell type hierarchy are desirable, the lower in the hierarchy we go, the less “independent” the cell types are (they will share more similar regulatory programs). When cell types are less “independent”, one would expect their marker genes to behave more similarly as well, thus leading to crowding of their marker genes (and thus mixing of colors in the feature embeddings).

Nonetheless, we identified matched cell types between CellTypist and the FetalLiverDataset based on name, then visualized the embeddings of marker genes for matching cell types. We assigned colors based on the lower hierarchy or the higher hierarchy from CellTypist (**Fig. S13**, see below). While clustering at lower hierarchy is harder to see due to co-clustering of marker genes from similar cell types, the clustering is more readily apparent in higher hierarchy with erythroid (green) clustering in upper right, macrophages in bottom left (red), and DC cells (orange) above the macrophages, and finally lymphocytes (blue) in upper left.

Fig. S13. Clustering of marker genes based on CellTypist markers requires higher hierarchy of cell types. Scatter plot shows the feature embeddings of siVAE and PCA trained on the fetal liver dataset, where each point represents a single input feature. **(a)** Scatterplot of siVAE feature embeddings, where colors are based on the higher hierarchy cell types. **(b)** Scatterplot of PCA loadings of genes, where colors are based on the higher hierarchy cell types. **(c)** Scatterplot of siVAE feature embeddings, where colors are based on the lower hierarchy cell types. **(d)** Scatterplot of PCA loadings of genes, where colors are based on the lower hierarchy cell types.

However, as is apparent from **Figure S13** both for siVAE factor loadings as well as e.g. PCA, separating marker genes of similar cell types is still overall challenging. **Figure S14** (below)

illustrates the overlap in marker gene sets between each of the cell types in the lower and higher hierarchy annotations of CellTypist, and you can see there's substantial overlap between the different cell types, supporting the idea that they are closely related, and many cell types share similar markers.

We found that the MSigDB marker sets (which in this specific case, are derived from a single cell atlas paper) for a subset of more-independent cell types led to better separation, which is why we chose to use them in our paper.

Fig. S14. Marker genes in CellTypist for different cell types overlap strongly. Heatmap indicates the overlap in marker gene set between cell types, when using CellTypist annotations at the (a) higher hierarchy or (b) lower hierarchy.

5. In the methods section titled "siVAE and VAE network design" it states: "Additional implementation details, as well as a table containing the above information on network design, can be found in the Supplementary Materials (Supplementary Table 1)." I wasn't able to find the additional implementation details, could you include a more specific pointer to where this is?

We have moved most of the implementation details to the main text, and corrected the above text to: "A summary of the network design can be found in **Supplementary Table 1**."

6. In Supplementary Table 1, I am unsure which latent dimensions were used for which experiment. This is clear for the fetal liver atlas, 2D for visualization and 64D for all other analyses. But what about the other datasets? Were these all used for a different experiment or only considered in the hyperparameter search for a good latent dimension?

For all other datasets (MNIST, Fashion-MNIST, CIFAR-10, 1.3 Million Brain cells), we experiment with multiple number of latent dimensions to show the effect of varying number of latent dimensions on performance. For imaging datasets, (MNIST, Fashion-MNIST, CIFAR-10), they also include additional analysis that were performed on latent embeddings of size 2, and this information has been added to the paper.

7. In addition, it's not readily apparent to a casual reader that in some cases two models are being trained - one for visualization and one for other downstream tasks. I think this should be mentioned in the main text and not only in the methods section.

We have explicitly added clarification on the result section: "Since direct visualization is not required for identification of hub gene, and we wanted to avoid restricting computational complexity of siVAE, we expanded the number of latent dimensions from 2, used for all visualization related analysis including clustering, to 64, for GCN related analysis."

8. In the methods section titled "Cell type classification." it states: "In addition, the number of clusters, was set to 15 as imaging datasets have far fewer classes than the fetal liver atlas dataset." I am confused by the notation here, isn't k the number of neighbors used, not the clusters?

We have modified our main text: "In addition, the number of neighbors, k , was set to 15 as imaging datasets have far fewer samples than the fetal liver atlas dataset."

9. Similarly, the same section states: "After training using the training fold, the encoders were used to compute embeddings for the training and test datasets. We then used a -NN ($k=80$) classifier to predict labels of test cells based on the embeddings of the training and testing datasets." I'm just a bit confused on the wording and want to make sure I got it correctly. So, after cross-validation was performed, you used your training set to get your final trained embedding. Using this trained embedding, you applied it to the combined training and test data sets. Then you learned a single kNN classifier on the embedded training data and evaluated the trained kNN model on the test data. Is this correct?

That is correct. We 1) generated both the train and test cell embeddings, 2) trained a kNN classifier on the train embeddings, then 3) applied the kNN classifier to the test embeddings.

10. Just to make sure I am understanding, for the fetal liver atlas, was the kNN classification done on the 64-dimensional latent embedding?

kNN was performed on the 2-dimensional latent embeddings to quantify how well cells cluster in the provided visualization. We have edited the main text to clarify this point.

11. In the method section titled "Execution time comparison experiments." It states "In the case of the feature attribution execution times, we extrapolated the execution time on the LargeBrainAtlas dataset from the execution time on 100,000 cells, due to time constraints and the fact that runtime of these methods should scale linearly with the number of cells. In contrast, execution times of siVAE are those measured on the full dataset." I think this is potentially misleading. I am naive, so I do not know if these "methods should scale linearly". I think this needs to be shown in some capacity, or at a minimum clearly stated in the figure caption.

We reasoned that since the attributions are calculated in batches of sizes much smaller than even 100,000 cells, and the computation time of each batch should be nearly identical, the runtime should scale linearly in this specific case. If there are additional overheads that accumulate

through batches, then our prediction would be underestimating the runtime of feature attribution methods (in their favor). To be sure, we plotted the individual execution time of 200 batches as a function of the order they were executed in, to show that runtime does not significantly differ across batches (mean=5.76 mins, std dev=0.35 mins) (**Fig. S10**). In addition, we modified the figure caption to indicate extrapolation was used for experiments with infeasibly long run time.

12. In the method section titled "Generating cell line embeddings with siVAE and graph kernel." it states "We then used in Weisfeiler-Lehman GraKeL111 to infer adjacency matrix per gene embedding then generate similarity matrix between inferred network generated per binned datasets." I would recommend rewording it a bit for people unfamiliar with GraKeL, it sounds like GraKeL only infers the adjacency matrix, then you generate a similarity matrix from it. But I think you mean to say that you generate the adjacency matrix from the gene embeddings then use GraKel to generate the similarity matrix.

Thank you for pointing this out. It has been corrected in the paper: "We then inferred the adjacency matrix from gene embeddings, then used Weisfeiler-Lehman GraKeL¹¹² to generate a similarity matrix for each network. The similarity matrices corresponding to the same cell line were averaged together before PCA visualization of the cell lines."

13. Page 12, line 24 it states "Surprisingly, we observed separation of cell lines according to their neuronal differentiation efficiency along PC-1, when visualizing all lines together (Fig. 6b), as well as when visualizing only iPSC lines that showed any differentiation (Fig. 12)." Can you calculate a correlation statistic here? PC-1 vs differentiation efficiency? Also I believe Supp Fig 12 should be 15.

We agree statistics is necessary here for reinforcing our statement. The correlation between PC-1 and differentiation efficiency for all FPP cell lines is (Spearman $\rho=0.62, P=3.0 \times 10^{-5}$), and the correlation for only differentiated FPP cell lines is (Spearman $\rho=0.59, P=4.8 \times 10^{-4}$), which are both significant and support our statement on relationship between the PC-1 of cell line embedding and differentiation efficiency. The statistics has been updated to the legend of **Figure S19**.

14. Page 12, line 32 it states "We confirmed on both the two progenitor cell types (P_FPP and FPP) that differentiation efficiency explains separation in the cell line embeddings (Fig. 15)," Is this from visual inspection? I am not sure I see this same separation -- can a correlation statistic be included here? I am most confused by the column "Diff. Cells Only", I don't see a clear separation, but maybe I am interpreting things incorrectly? Also, S15 is missing a color key.

All plots show significant correlation between PC-1 and differentiation efficiency:

- all cell lines with cell type FPP (Spearman $\rho=0.62, P=3.0 \times 10^{-5}$)
- differentiated cell lines with cell type FPP (Spearman $\rho=0.59, P=4.8 \times 10^{-4}$)
- all cell lines with cell type P_FPP (Spearman $\rho=0.55, P=4.2 \times 10^{-3}$)
- differentiated cell lines with cell type P_FPP (Spearman $\rho=0.53, P=0.019$).

The result has been added to the legend of **Figure S19**.

15. Line 49 page 6 it says "We tested our model on an iPSC neuronal differentiation (NeurDiff) dataset in which XX cell iPSC..." what is XX?

Thank you for pointing this out, XX has been replaced by 253,381 for number of iPSC-derived cells.

16. TP10K is used twice in the methods but is not explained. It would be nice to write it out once for readers unfamiliar with this acronym.

We have added transcripts per 10k transcripts (TP10K) in the paper for clarification: "We normalized the count matrix to TP10K (number of transcripts per 10K transcripts), then performed feature selection by retaining the top 2,000 highly variable genes, yielding 177,376 cells and 2,000 genes."

Major comments:

1. The proposed method has a lot of resemblance of scETM (Zhao et al., Nat Comm 2021) (source code: <https://github.com/hui2000ji/scETM>) except for a few technical differences. Although authors cited the paper, they did not perform thorough comparison with scETM as they did with LDVAE. I suggest that they compare siVAE with scETM in both cell type classification (Figure 2) and query gene prediction (Figure 5).

We initially did not perform the experiments on scETM as we viewed that its performance would still be limited by the ultimately linear decoder (which we tried to avoid), as is the case for LDVAE. To address the reviewer's concern, we performed the requested comparison. We first tested with their default settings while matching the encoder architecture of siVAE. We also performed hyperparameter tuning, where we varied combination of select hyperparameters (learning rate, KL weight, # topics), then selected the best model. Regardless of the hyperparameters selected for scETM, its performance remained below even LDVAE.

The results for scETM have been added to **Figure 2b** and **Figure 5c** (see below).

2. The feature encoder size grows as the number of training cells, which is a main issue when dealing with millions of cells (e.g., LargeBrainAtlas). The authors remedied this issue by either downsampling cells or applying PCA to get C_red eigenvector PC cells. However, it is not clear to me how the feature encoder applies to test cells. My guess is the feature encoder is discarded and only the cell encoder is used to generate embedding for a test cell, which is the same as the standard VAEs. Therefore, the feature encoder is not transferable. Please verify.

The reviewer is correct, only the feature decoder and cell-wise encoder-decoder is used at test time. To clarify:

- During training, scVAE only sees the training cells.
- The parameters of both the cell-wise and feature-wise encoder-decoders are fit during training
- After training, every feature has an embedding calculated from the entire training dataset; the feature embeddings are fixed after training and do not change.

- During testing, the test cells are only put through the cell-wise encoder to generate test cell embeddings.
 - For reconstruction of the test cell, the test cell embedding is combined with the feature embeddings to reconstruct the input.
3. Since the feature embedding and cell embedding are both unbounded real value with no L1 sparsity regularization or non-negative constraint, interpreting feature embedding based on whether it's highly positive (i.e., as illustrated in Figure 1d) can be misleading. The problem is that a highly positive values in the score embedding space can be canceled out by the high negative values in the loading embedding space when performing the reconstruction so the net result is the same as having zero score and loading. This is the reason why non-negative matrix factorization or topic models are usually preferred choices to interpret learned latent factors over PCA or probabilistic PCA. Can authors comment on that?

We agree in the case of PCA, there can be canceling of large positive and large negative values. In the case of siVAE however, the linear terms involving the factors and scores only plays a minor role in the regularization term (which is specifically weighted to be small, and used primarily to encourage siVAE to learn correspondence between the dimensions of the feature and cell embeddings). siVAE at its core is still a VAE, and by virtue of the non-linear use of the embedding dimensions to reconstruct the original expression profile, in practice it would be challenging to identify the cases in which cancellation would occur.

We therefore relied on our comparisons against the neural network feature attribution methods to confirm our interpretation of the embeddings (that large values indicate higher contribution of a feature to variation along a dimension). **Figure 3** shows high agreement between siVAE, DeepLIFT and grad*input.

To a somewhat less relevant extent, we also want to highlight our observation that genes farther away from the origin (in the embedding space) have both lower reconstruction error (**Fig. S11**) and therefore higher degree centrality (**Fig. S15**) in the underlying co-expression network, supporting the idea that distance from the origin is correlated with some measure of 'importance' of the gene.

4. The authors used KNN for cell type classification. Adjusted Rand Index (ARI) is also a commonly used metric especially for evaluating unsupervised clustering methods where the cell types are the ground truth cluster labels and the Louvain or Leiden cluster indices using the model-dependent embedding are the predicted clusters. Please consider adding ARI as another metric when comparing with the existing methods.

We agree with both reviewers on this point. We have calculated both balanced classification accuracy as well as ARI, and have found that the results are consistent across different metrics with siVAE outperforming other methods (besides the VAE), and rankings of performance are consistent across metrics (**Fig. 2b, S2, S3**). For example, see **Fig. S2** reproduced below.

5. Authors experimented Negative Binomial as the data likelihood siVAE (NB) as opposed to Gaussian likelihood used in siVAE and saw worse performance. Can authors clarify whether their siVAE (NB) operates on log-transformed data or raw count data?

siVAE (NB) as well as any other model using the NB distribution is operating on raw counts. We have now clarified this in the text: “For model configurations of LDVAE and scVI that use negative binomial distribution for modeling counts, raw counts were used as observations.” and “For both siVAE variants that use the negative binomial distribution for modeling counts, raw counts were used as observations.”

6. The authors defined hub genes as the genes with high reconstruction accuracy compared to other genes. However, housekeeping genes may also be well predicted since their expression don't change much but they are often not hub genes. Can author explain that?

During pre-processing, we perform feature selection to pick highly variable genes (e.g. the top 2000 most variable genes), as is standard for single cell data processing. We reason that housekeeping genes are typically less variant (as the reviewer suggests) and so are likely to be removed during this step.

Furthermore, under the non-NB versions of siVAE, all input genes are centered and scaled during preprocessing. For genes that are consistently expressed across samples, once you remove their mean expression (through centering), their remaining variation behaves more like noise, and so we would be less likely to observe correlation between e.g. housekeeping genes. Because siVAE can only drive down reconstruction error by capturing co-expression patterns between features, it would not be able to drive down reconstruction error of the housekeeping genes.

7. Related to comment 6, are the identified hub genes TF master regulators? Can authors show that the gene co-expression network inferred by siVAE recovers any known regulatory or interaction network information such as TFBS, PPI, chromatin co-accessibility, etc?

It is difficult to use recovery of known hubs of a regulatory network as validation; the reason is that what is considered a ‘hub’ by siVAE (as well as any other dimensionality reduction or co-expression network inference method) depends on the input dataset.

To illustrate this, consider a dataset consisting of K cell types, with each cell type contributing the same number of cells, and where most of the variation in expression is observed between cell types (e.g. relatively little intra-cell type variation). In this scenario, there would be many

differentially expressed genes (because most of the variation in the dataset is inter-cell type); let's presume that it would be relatively easy to find a number of 'marker genes' that are more highly expressed in each cell type than the other $K-1$ cell types. If we trained siVAE (or another method) on this dataset, siVAE would learn to use those marker genes to essentially predict the rest of the genome; intuitively, the marker genes would be able to label a given cell in terms of its type, then during model training, the model would just need to roughly "memorize" an exemplar from that cell type and learn to output that exemplar, in order to achieve low reconstruction error. siVAE would therefore reconstruct test cells fairly well, and the 'hubs' it learns would be the marker genes. In this instance, siVAE is capturing "covariation" of marker genes with all other genes that are differentially expressed in a cell type versus the others.

If we then instead trained siVAE on each of the individual K cell types separately, the marker genes it used in the above scenario would no longer be good predictors of expression within each cell type. The reason is that within a single cell type, the marker genes for that cell type are likely less variant, so the model would look for different genes that predict co-expression within the cell type. These co-varying genes it focuses on would likely be very different from the co-expressed genes it picks up in the above scenario.

Specifically with respect to being able to pull down a transcription factor and its native targets in the genome, this can be a tricky process. To pull down a single edge between a TF and its target, one has to have cells (or samples) in which the TF is highly expressed (and leads to high expression of the target), but also have samples where the TF is poorly expressed (and which leads to poor expression of the target). Thus, if in a particular cell type a TF is typically 'on' (and therefore the target is also always 'on' in the cells), after centering the genes, there's no more coexpression of the genes in the samples (this is similar to the argument above about capturing housekeeping genes). In reality, you would need to include cell types in which a factor is on and upregulating its target, but also less related cells (or the same cells in a different context) in which the factor is off, so you can also observe the target gene is off as well. In short, there must be strong domain knowledge about where and when particular factors and their targets are expressed AND not expressed, in order to capture those interactions in a co-expression network.

This is why in our network experiments, we choose to establish a 'ground truth' degree centrality of a node in the network, and compare siVAE's degree centrality against that measure, in order to establish its accuracy (because the ground truth set of hubs can be independently established for any given input dataset, whereas if we looked for a database of known TF-target interactions, we may not be able to identify data and the contexts in which the interaction does and does not happen).

8. The simulation presented in Figure 4a,b and described in the main text is a little too simplistic. It's a toy data for sanity check. Author should consider moving it to supplementary (as they did for MNIST).

We agree simulation is simplistic, but it serves as a necessary introduction to the concept of co-expressed genes co-localizing in the feature embedding space in the case where we have a ground truth co-expression network (which is not true for the real dataset experiments). Therefore we have ultimately decided to keep the section in the main text.

9. The description for the generative processes of both VAE and siVAE are not accurate. Starting with the standard VAE, the generative process does not involve any neural network, i.e., $z_c \sim N(0, I_K)$; $X_c \sim N(\theta z_c, \sigma I)$. It's the variational inference that involves it i.e., $q(z_c|X_c)$

$\sim N(\mu(X_c), \sigma(X_c))$. Similarly, the generative process of the siVAE should be: (a) $z_c \sim N(0, I_K)$; (b) $v_f \sim N(0, I_K)$; (c) $X_{c,f} \sim N(z_c v_f, \sigma)$. It's the variational inference involves network encoders $q(z_c|X_c)$ and $q(v_f|X_f)$. Having a neural network in the generative process itself will lead to non-closed-form ELBO since the prior distribution of the network-transformed Gaussian (i.e., $f(z_c, l)$ for $l > 1$) is not a Gaussian.

The VAE (and siVAE, by extension) actually does use a non-linear decoder (defined by a neural network); we refer the reviewer to the original VAE paper by Kingma (Autoencoding Bayes, 2013, Kingma, Appendix C.2), where the decoder network with one hidden layer is defined as $X \sim N(\mu, \sigma^2 I)$ and $\mu = W_4 h + b_4$, with h as the hidden layer ($h = \tanh(W_3 z + b_3)$). More complex VAEs can be defined by including more hidden layers, changing the activation function, etc. If it was only the encoder that uses a neural network, but with a linear decoder, then we would be performing variational inference on a linear dimensionality reduction method (like LDVAE).

10. The authors mentioned that they focused on latent space of the first layer of both cell encoder (i.e., siVAE scores) and feature encoder ($l=1$) (i.e., siVAE loadings), which means the purpose of the deep layers $l > 1$ are mainly for better reconstruction. However, in standard VAE, it's the output from the last layer L (i.e., $[z_{1,L}, \dots, z_{C,L}]$ and $[v_{1,L}, \dots, v_{F,L}]$ in the context of siVAE) that are used for downstream analysis. The rationale of using the first layer $l=1$ is unclear.

There may be a misunderstanding with the naming scheme. In short, our analyses generally focus on the (inferred) latent embedding of the cell ($z_{c,1}$, the output of the encoder); the other variables $z_{c,\ell}$ for $\ell > 1$ represent the activations of the hidden layer ℓ of the decoder for cell c , and are transformations of the latent embedding (but are not technically the latent embedding). We've clarified this in the main text and in **Fig. S23** shown below: " $z_{c,1}$ is the embedding of cell c in the (latent) cell embedding space of the VAE, while $z_{c,\ell}$ for $\ell > 1$ represent the activations of the hidden layer ℓ of the decoder for cell c ."

11. In Equation 12 for the ELBO (which in its correct form should be: $ELBO = E[\log p(X|z) + \log p(z) - \log q(z)] = E[\log p(X|z)] - E[\log q(z) - \log p(z)] = E[\log p(X|z)] - KL[q(z)||p(z)]$), the minus sign is missing in front of I_{KL} (i.e., $E[\log q - \log p]$). Same error in Equation 18 and 20. Also, equations 13 and 23 (i.e., the interpretability term) should have a plus sign instead of minus sign (since you want to encourage linear reconstruction for interpretation). I am not sure whether this is a

conceptual understanding mistake or the authors in fact implemented the wrong objective functions the same way as they described.

Thank you for pointing out the error, we mixed up the signs in the text but not the implementation, which we have confirmed. We have corrected the equations in the text, reproduced below.

We perform variational inference and learning by maximizing the expected lower bound function

ℓ_{SIVAE} , where $\ell_{\text{KL}} = \text{KL}\left(q\left(\{\mathbf{v}_{f,1}\}_{f=1}^F, \{\mathbf{z}_{c,1}\}_{c=1}^C\right) \parallel p\left(\{\mathbf{v}_{f,1}\}_{f=1}^F, \{\mathbf{z}_{c,1}\}_{c=1}^C\right)\right)$.

$$\ell_{\text{SIVAE}} = -\ell_{\text{KL}} + \mathbb{E}_{q(\mathbf{z}_{c,1}, \mathbf{v}_{f,1})} \left[\sum_c \sum_f \log N(X_{c,f}; \mathbf{v}_{f,1}^T \mathbf{z}_{c,1}, \sigma_d(\mathbf{z}_{c,1})) \right] \quad 12$$

$$+ \gamma \mathbb{E}_{q(\mathbf{z}_{c,1}, \mathbf{v}_{f,1})} \left[\sum_c \sum_f \log N(X_{c,f}; \mathbf{v}_{f,1}^T \mathbf{z}_{c,1}, 1) \right] \quad 13$$

$$\ell_{\text{SIVAE}_{\text{NB}}} = -\ell_{\text{KL}} + \mathbb{E}_{q(\mathbf{z}_{c,1}, \mathbf{v}_{f,1})} \left[\sum_c \sum_f \log \text{Poisson}(X_{c,f}; l_c m_{c,f}) \right] \quad 18$$

$$X_{c,f} \sim N(\mathbf{v}_{f,1}^T \mathbf{z}_{c,1}, \sigma_d(\mathbf{z}_{c,1})) \quad 19$$

$$\ell_{\text{SIVAE}_{\text{LINEAR}}} = -\ell_{\text{KL}} + \mathbb{E}_{q(\mathbf{z}_{c,1}, \mathbf{v}_{f,1})} \left[\sum_c \sum_f \log N(X_{c,f}; \mathbf{v}_{f,1}^T \mathbf{z}_{c,1}, \sigma_d(\mathbf{z}_{c,1})) \right] \quad 20$$

$$\ell_{\text{SIVAE}} = \ell_{\text{KL}} + \mathbb{E}_{q(\mathbf{z}_{c,1}, \mathbf{v}_{f,1}, s_c)} \left[\sum_c \sum_f \log N(X_{c,f}; \mathbf{v}_{f,1}^T \mathbf{z}_{c,1}, \sigma_d(\mathbf{z}_{c,1})) \right] \quad 22$$

$$- \gamma \mathbb{E}_{q(\mathbf{z}_{c,1}, \mathbf{v}_{f,1})} \left[\sum_c \sum_f \log N(X_{c,f}; \mathbf{v}_{f,1}^T \mathbf{z}_{c,1} + j_f^T s_c, 1) \right] \quad 23$$

Minor comments:

1. I suggest avoid using GCN as its connotation is Graph Convolutional Network in the deep learning community. I personally found it quite confusing since GCN has been used throughout the Methods section that I started reading before the Results section. Maybe GCEN is a better abbreviation.

Thank you for the suggestion. It is highly unfortunate that the same acronym is common in both the biology and deep learning communities and have different meanings. Since we are aiming this paper at the biology network community more than the deep learning community, we have consciously made the decision to stick with the GCN term; otherwise the biology community may be confused and think we are referring to another concept.

2. The paper needs careful proofreading. Page 22 line 35, "The five-fold nested cross validation experiments reported in Figure 1c", I think "Figure 1c" is the wrong reference

The typo has been fixed in the paper: “The five-fold nested cross validation experiments reported in **Figure 2b** compares the performance of siVAE, VAE, and LDVAE on the fetal liver atlas dataset when matching their cell-wise encoder and decoder network designs.”

3. Figure 1 legend "Note the embeddings of genes 1, 2, 3 and 4 all have large magnitudes along dimension 1 but not dimension 2," should be describing panel d not panel b.

The typo has been fixed in the paper: “Note in (d) the embeddings of genes 1, 2, 3 and 4 all have large magnitudes along dimension 1 but not dimension 2, suggesting genes 1, 2, 3 and 4 explain variation in the cell embedding space along dimension 1. Genes 5, 6, 7, and 8 sit at the origin of the feature embedding space, suggesting they do not co-vary with other features.”

4. Figure 2 "embeddfing" => embedding

The typo has been fixed in the paper: “2D visualization of the inferred cell embedding space using UMAP and siVAE with batch correction.”

Reviewer #3

1. In the title they use genomics, but it is not tested on genomic associations, which is misleading. Better to use siRNA-seq in title as well.

While siVAE could be applicable to other types of genomic datasets, we agree the primary focus of this paper is on scRNA-seq, and have modified our title to “Interpretable deep generative models for single cell transcriptomics.”

2. Authors spend substantial time comparing it to other methods, which is good, However they don't show so much external biological validations. An exception is the pathway associations and neuronal differentiation, which I liked. However, the role of hubs are key in complex diseases and cancers. Given the great variety of readers in Genome Biol I believe a sc data-sets and topological analysis in such a data-set would be showing how the method could be used for precisions medicine applications.

Thank you for the suggestion to showcase wider applicability of siVAE. We have performed an additional network-based analysis of a subset of the SEA-AD: Seattle Alzheimer's Disease Brain Cell Atlas dataset and added the result to the main text. The dataset is comprised of single cells sequenced from cases (dementia) and controls from multiple donors across 5 cell types (L2/3 neuron, L4 neuron, L5 neuron, oligodendrocyte, Vip). We applied siVAE to identify subnetworks associated with dementia. In summary, we identified a strong dementia effect in the global gene networks of oligodendrocytes and Vip neurons (**Fig. 6e**). Upon closer analysis of the network changes associated with dementia, we identified UBB, whose frameshift mutant UBB+ is known as strong hallmark of Alzheimer's, as a hub gene (**Fig. 6f**). We observed that UBB loses significant coexpression patterns in individuals with dementia. We reason that these results highlight the potential for siVAE to help identify network structural features associated with clinical phenotypes (e.g. dementia status) that may indicate a role for changes in co-expression in disease etiology. Please find below a cropped version of **Figure 6** containing the relevant figure for this analysis (**Fig. 6e,f**).

3. It would be interesting if authors discussed how one could take this to another level using multi-omics siRNA-seq data. It feels like this could be very good data for this model associating for example ATAC-seq and RNA-seq. This could be done in a forthcoming paper.

Thank you for the suggestion, we have been considering this cross-omics application and added possibility of this future work in the discussion.

“A promising application of siVAE is in multimodal data analysis. Assays such as SNARE-Seq⁷¹ that jointly measure gene expression and chromatin accessibility from the same cell, or CITE-Seq⁹⁶ that jointly measures gene expression and protein expression, are useful for characterizing how variation in one modality (e.g. RNA) relate to variation in another modality (e.g. chromatin accessibility). For VAE-based multi-modal analysis methods (totalVI⁹⁷, multiVI⁹⁸, scMVP⁹⁹, BABEL¹⁰⁰, and Cobolt¹⁰¹), the siVAE interpretability term could be easily incorporated to identify how features from one modality map to another, such as linking enhancers based on chromatin accessibility to its target genes.”

4. A minor comment is that figure 2 says that $\gamma=0.05$ was used, but other places indicate $\gamma=0$. Seems to be a typo there.

We have modified **Figure 2a** to clarify which setting was used (see below).

a

Second round of review

Reviewer 1

Huge thanks to the authors who thoroughly and methodically responded to each of my questions and concerns. I only have three small points:

1) Thanks for the clarification and the additional analysis using lower- and higher-resolution cell types from CellTypist. Figures S13 and S14 drive home the point that clustering input features are inherently difficult to do when considering the high overlap of genesets. It also shows that you aren't cherry-picking specific results in Figure 4c, but that you are justified in highlighting those 4 cell types. One small suggestion, I don't see NK/NKT cells or hepatocytes in figures S13/S14. It would be useful to visualize how different the 4 cell types you highlight differ in their gene sets in comparison to all the other cell types of interest using your MSigDB genesets, similarly to how you have done it in S14.

2) Thanks for the addition of Figure S3. It's pretty interesting to me how some methods perform much better in some cell types than others. Small edit: Figure S3 panel Pre-B cell still has the legend.

3) Thank you for your additional experiment and Figure S10. I am convinced by your arguments. However, I don't see the change in the caption for Figure 3b.

Reviewer 2

Thank you for addressing my comments. I have a few additional comments.

1. In describing LDVAE, scVI, and scETM, the authors repeated this sentence "The architecture of the model was set to match that of the cell-wise encoder-decoder of siVAE, including the number of dimensions of the cell embedding space and the number of hidden layers, as well as the number of hidden nodes". This may not be fair for those methods since the setting chosen was fine-tuned for siVAE. While they specified the learning rate, KL weight, and number of topics, the embedding size was not described. From the "siVAE and VAE network design" section, they mentioned embedding size of 2, 5, 10, 20, which are all quite small. Please specify clearly the embedding size for the former 3 existing methods so that the comparison is fair.

2. Also, the number of epochs to train LDVAE, scVI, and scETM need to be specified. In particular, did the authors make sure that they converge before evaluating their performance?

3. Please add scETM to Table 1 (since it's used as a comparison method now).

4. Equation 22 is still not correct. Namely, the minus sign is missing in front of l_{KL} and the linear reconstruction should have had plus sign instead of minus sign.

Authors Response

We thank the reviewers for their helpful feedback and constructive criticism. Below, we address their concerns:

Reviewer #1

1) Thanks for the clarification and the additional analysis using lower- and higher-resolution cell types from CellTypist. Figures S13 and S14 drive home the point that clustering input features are inherently difficult to do when considering the high overlap of genesets. It also shows that you aren't cherry-picking specific results in Figure 4c, but that you are justified in highlighting those 4 cell types. One small suggestion, I don't see NK/NKT cells or hepatocytes in figures S13/S14. It would be useful to visualize how different the 4 cell types you highlight differ in their gene sets in comparison to all the other cell types of interest using your MSigDB genesets, similarly to how you have done it in S14.

NK/NKT cells and hepatocytes were not included in the S13/S14 figures because CellTypist did not contain marker gene sets for those two cell types at the higher hierarchy level.

We have now added a comparison of overlap between the 4 cell types available from MSigDB and CellTypist (**Figure S14c**). As expected, related cell types across MSigDB and CellTypist show the most overlap in genes (i.e. the pair of cell types with largest overlap across MSigDB and CellTypist are CellTypist/lymphocytes and MSigDB/B cells, followed by CellTypist/lymphocytes and MSigDB/NK/NKT cells). Overall, it is clear that the 4 chosen cell types from MSigDB have no overlap in genes, while CellTypist gene sets show significant overlap between themselves, which is consistent with the clear separation of MSigDB gene sets in siVAE gene embedding space. Shown below is the added **Figure S14c**:

We have added the following text into the manuscript: “Note the gene sets of similar cell types across MSigDB and CellTypist (higher hierarchy) are consistent (**Fig. S14c**).”

2) Thanks for the addition of Figure S3. It's pretty interesting to me how some methods perform much better in some cell types than others. Small edit: Figure S3 panel Pre-B cell still has the legend.

We have corrected this (see below).

3) Thank you for your additional experiment and Figure S10. I am convinced by your arguments. However, I don't see the change in the caption for Figure 3b.

We have modified the **Figure 3b** caption as follows: "Bar plot indicating the time required to train siVAE versus training a classic VAE and applying feature attribution methods on the LargeBrainAtlas dataset. For feature attribution methods, the run times were extrapolated due to infeasibly long run times; additional experiments demonstrate the feasibility of extrapolation (**Figure S10**)."

Major comments

1. In describing LDVAE, scVI, and scETM, the authors repeated this sentence "The architecture of the model was set to match that of the cell-wise encoder-decoder of siVAE, including the number of dimensions of the cell embedding space and the number of hidden layers, as well as the number of hidden nodes". This may not be fair for those methods since the setting chosen was fine-tuned for siVAE. While they specified the learning rate, KL weight, and number of topics, the embedding size was not described. From the "siVAE and VAE network design" section, they mentioned embedding size of 2, 5, 10, 20, which are all quite small. Please specify clearly the embedding size for the former 3 existing methods so that the comparison is fair.

The embedding sizes for the 3 former methods are all set to 2, which is consistent with the embedding size for siVAE in the benchmarking experiment with the fetal liver atlas dataset. The embedding size of 2, 5, 10, 20 was used for imaging datasets where we tested the effect of embedding size on siVAE performance, with results shown in **Figure S7** (reproduced below).

As can be seen above, siVAE also experiences significant performance loss with only 2 embedding dimensions. However, the embedding size was purposefully set to be small so that the embedding space can be directly visualized without further dimensionality reduction methods such as t-SNE or UMAP, which would prohibit direct interpretation of the 2-dimensional space. Since all methods are limited by the same embedding size, we believe the comparison is fair. We have clarified in the text that we set every method's embedding size to 2: "We compared siVAE against a classic VAE as well as LDVAE and scETM, where all four VAE frameworks used cell-wise encoder-decoders of the same size with latent dimension of 2, and the VAE and siVAE use the same activation functions" and "The architecture of the model was set to match that of the cell-wise encoder-decoder of siVAE with latent dimension of 2."

2. Also, the number of epochs to train LDVAE, scVI, and scETM need to be specified. In particular, did the authors make sure that they converge before evaluating their performance?

For LDVAE and scVI, we used 100 epochs and visually confirmed from the train/test loss graph that the models are converged. We have added the graphs demonstrating convergence as **Figure S25**, reproduced below.

“Fig. S25. Train/test losses for LDVAE and scVI. Line plot of train and test losses for LDVAE and scVI, demonstrating convergence of the models.”

For scETM, their documentation did not provide access to the train/test loss history, but we trained the model between 50, 100, 500 epochs and chose the saved model with lowest loss, and have updated this to the main text: “Per configuration, we trained the model for varying number of epochs {50,100,500} and chose the trained state with lowest loss”

3. Please add scETM to Table 1 (since it's used as a comparison method now).

Table 1 has been updated to include scETM.

Model	Interpretability Reg.	Decoder	Observation model
siVAE	Yes	Non-Linear	Gaussian
siVAE ($\gamma = 0$)	No	Non-linear	Gaussian
siVAE-linear	No	Linear	Gaussian
siVAE-NB	Yes	Non-linear	Negative Binomial
VAE	NA	Non-linear	Gaussian
scVI	NA	Non-linear	Negative Binomial
LDVAE	NA	Linear, single	Negative Binomial
scETM	NA	Linear, tri-factorization	Gaussian

VAE (linear)	NA	Linear, single	Negative Binomial
--------------	----	----------------	-------------------

4. Equation 22 is still not correct. Namely, the minus sign is missing in front of ℓ_{KL} and the linear reconstruction should have had plus sign instead of minus sign.

Thank you for pointing this out, the error has been corrected in the main text (see below)

$$\ell_{\text{SIVAE}} = -\ell_{\text{KL}} + \mathbb{E}_{q(\mathbf{z}_{c,1}, \mathbf{v}_{f,1}, s_c)} \left[\sum_c \sum_f \log N(X_{c,f}; \mathbf{v}_{f,L}^T \mathbf{z}_{c,L}, \sigma_d(\mathbf{z}_{c,L})) \right] \quad 22$$

$$+ \gamma \mathbb{E}_{q(\mathbf{z}_{c,1}, \mathbf{v}_{f,1})} \left[\sum_c \sum_f \log N(X_{c,f}; \mathbf{v}_{f,1}^T \mathbf{z}_{c,1} + j_f^T s_c, 1) \right] \quad 23$$